# DEL-Ranking: Ranking-Correction Denoising Framework for Elucidating Molecular Affinities in DNA-Encoded Libraries

## Abstract

DNA-encoded library (DEL) screening has revolutionized protein–ligand binding detection by enabling efficient exploration of vast chemical spaces through read count analysis. Despite its transformative potential, two critical challenges limit its effectiveness: (1) stochastic noise in low copy number regimes, where Poisson fluctuations significantly distort binding signals, and (2) systematic biases between observed read counts and actual binding affinities due to experimental artifacts and amplification variability. We introduce DEL-Ranking, a comprehensive framework that addresses these dual challenges through targeted innovations. To mitigate stochastic noise, we incorporate a dual-perspective ranking mechanism that prioritizes stable relative ordering relationships over volatile absolute counts. To bridge the read count-affinity gap, our Chemical-Referenced Correction (CRC) module identifies critical binding-related functional groups and leverages these structure-activity insights to guide precise count adjustments. A key contribution is our release of three novel DEL datasets featuring 2D molecular sequences, 3D conformational data, and functionally-derived activity labels—addressing a significant resource gap in the field and enabling more robust method development. Rigorous validation across multiple datasets reveals that DEL-Ranking consistently outperforms existing methods, achieving a remarkable 28% improvement in Spearman correlation even under high-noise conditions. Our framework both enhances identification of high-affinity compounds and reveals novel functional motifs–Pyrimidine Sulfonamide, beyond known Benzene Sulfonamide groups. These interpretable insights accelerate therapeutic candidate discovery while advancing understanding of molecular recognition mechanisms.

## 1 Introduction

DNA-encoded library (DEL) technology has revolutionized protein-ligand binding detection by enabling parallel screening of vast compound collections against biological targets (Franzini et al., 2014; Neri & Lerner, 2018; Peterson & Liu, 2023; Ma et al., 2023). Unlike traditional high-throughput methods that screen compounds individually, DEL technology links each small molecule to a unique DNA barcode, allowing simultaneous evaluation of billions of compounds in a single experiment (Brenner & Lerner, 1992; Goodnow Jr & Davie, 2017; Yuen & Franzini, 2017). The DEL screening process (shown in Figure 1) generates read count data that serves as a proxy for binding affinity (Machutta et al., 2017; Foley et al., 2021). Specifically, these read counts represent the frequency of each compound detected after target binding and processing, with higher counts generally suggesting stronger binding. Experiments typically generate two types of counts: matrix counts (from control samples without target protein) and target counts (from samples with the target protein) (Favalli et al., 2018).

Despite DEL's potential for accelerating drug discovery (Satz et al., 2022; Neri & Lerner, 2017), two fundamental challenges limit its effectiveness and accuracy: 1) **Distribution Noise**: Read counts are highly variable, especially for compounds with few copies in the library. These compounds are subject to significant Poisson statistical fluctuations, distorting the relationship between counts and actual binding properties (Kuai et al., 2018; Favalli et al., 2018). 2) **Distribution Shift**: Systematic biases exist between observed read counts and actual binding affinities due to factors including

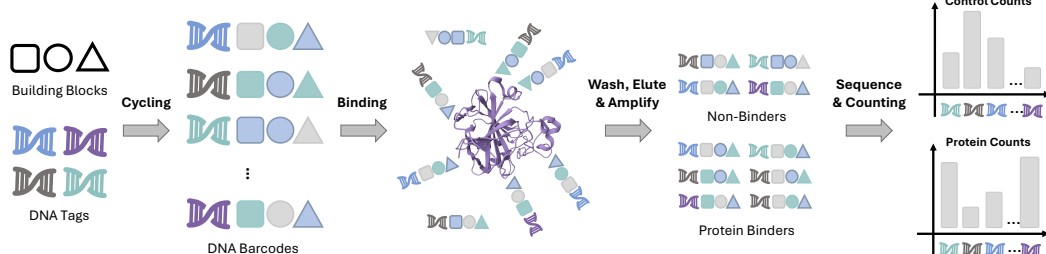

Figure 1: Illustration of the DEL screening process. **Cycling**: Creating unique compounds, each tagged with a distinctive DNA sequence. **Binding**: These compounds are then exposed to the target protein. **Wash, Elute and Amplify**: Compounds that bind to the target are retained, while others are washed away. The DNA tags of the bound compounds are then amplified and analyzed using sequencing techniques. **Sequence & Counting**: This process results in a distribution of read counts for target-bound samples and control samples.

synthesis efficiency and amplification variability (Yung-Chi & Prusoff, 1973; Kuai et al., 2018). This creates a fundamental gap between enrichment measurements and true binding strength.

Early computational approaches addressed these challenges through threshold-based filtering of enrichment factors (target/matrix count ratios) (Gu et al., 2008; Kuai et al., 2018). While computationally efficient, these methods ignored molecular structure information. More recent machine learning approaches captured non-linear relationships between molecular structures and count data (McCloskey et al., 2020; Ma et al., 2021), later enhanced with distribution constraints and molecular embeddings (Lim et al., 2022; Hou et al., 2023). DEL-Dock (Shmilovich et al., 2023) further improved performance by incorporating 3D conformational information with Zero-Inflated Poisson (ZIP) modeling.

Despite these advances, limitations persist. Current methods focus predominantly on absolute read count values rather than more stable relative rankings. Additionally, while certain **functional groups** correlate strongly with binding activity (Hou et al., 2023; Blevins et al., 2024), existing systems underutilize these structure-activity relationships (Wichert et al., 2024). To overcome these limitations, we propose DEL-Ranking, a comprehensive framework with several key contributions:

- **Novel Methodology**: Our approach addresses both challenges through complementary innovations: (1) a Dual-Perspective Ranking Strategy that mitigates Distribution Noise by prioritizing stable relative ordering over volatile absolute counts through Pair-wise Soft Rank (PSR) and List-wise Global Rank (LGR) constraints; and (2) a Chemical-Referenced Correction (CRC) module that addresses Distribution Shift by leveraging **functional group information as binary labels** to bridge the gap between read counts and binding affinities.

- **Comprehensive Datasets**: We release three novel DEL datasets that address a critical resource gap in the field. Unlike existing public DEL datasets that typically contain only molecular structures and read counts, our datasets uniquely combine 2D molecular sequences, 3D conformational data, read counts, and–critically–binary affinity labels derived from functional group analysis. These comprehensive resources provide the research community with multi-target datasets that enable more robust method development and validation.

- **Validated Performance**: Experiments across five diverse DEL datasets demonstrate consistent improvements over state-of-the-art methods, including a 28% increase in Spearman correlation under high-noise conditions. Our framework not only enhances identification of high-affinity compounds but also reveals novel binding-relevant functional motifs, such as Pyrimidine Sulfonamide groups, extending beyond the established Benzene Sulfonamide structures previously known to correlate with binding activity.

## 2 RELATED WORKS

**Traditional Approaches** include QSAR models (Martin et al., 2017) and molecular docking simulations (Jiang et al., 2015; Wang et al., 2015), which offer interpretability and mechanistic insights

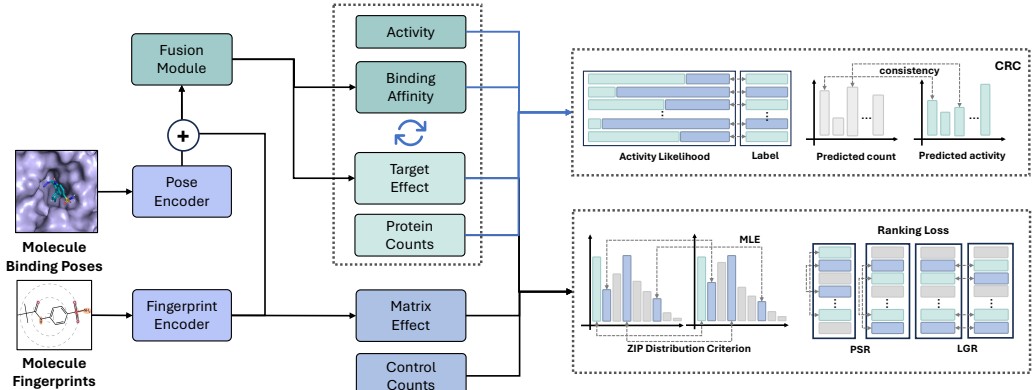

Figure 2: Overview of DEL-Ranking framework. The model directly fuses molecule binding poses and fingerprints as input features. CRC employs target effects and binding affinity to enhance read count prediction. The ranking-based loss incorporates target effects and matrix effects for noise removal, improving the correlation between predicted read counts and true binding affinities.

into protein-ligand interactions. DEL-specific techniques such as data aggregation (Satz, 2016) and normalized z-score metrics (Faver et al., 2019) were developed to address the unique challenges of DEL screening data. Despite their theoretical foundations, these approaches often struggle with scalability and capturing complex, non-linear relationships in large-scale DEL datasets.

**Machine Learning Methods** including Random Forest, Gradient Boosting Models, and Support Vector Machines, were used to improve DEL data analysis (Li et al., 2018; Ballester & Mitchell, 2010). These approaches, particularly when combined with Bayesian Optimization (Hernández-Lobato et al., 2017), offer enhanced scalability and better capture the non-linear relationships in high-dimensional DEL data. While outperforming traditional methods, they remain limited by their dependence on extensive training data and lack of interpretability when modeling complex biochemical systems.

**Deep Learning Approaches**, especially Graph Neural Networks (GNNs), have significantly advanced protein-ligand interaction predictions in DEL screening. GNN-based models effectively predict enrichment scores while accommodating technical variations (Stokes et al., 2020; Ma et al., 2021), and Graph Convolutional Neural Networks (GCNNs) enhance detection of complex molecular substructures (McCloskey et al., 2020; Hou et al., 2023). Recent innovations include DEL-Dock (Shmilovich et al., 2023), which combines 3D conformational information with 2D molecular fingerprints, address noise from truncated library products and sequencing errors (Kómár & Kalinic, 2020). Large-scale prospective studies have validated these AI-driven approaches, confirming improved hit rates and specific inhibitory activities against protein targets (Gu et al., 2024).

## 3 METHOD

To address **Distribution Noise** and **Distribution Shift**, we present DEL-Ranking framework by directly denoising read count values and incorporating novel activity information. 3.1 formulates the DEL denoising task; 3.2 and 3.3 introduce our innovative modules to address **Distribution Noise** and **Distribution Shift**; 3.4 and 3.5 introduce the overall training objective and framework architectures.

### 3.1 PROBLEM FORMULATION AND PRELIMINARIES

**DEL Prediction Framework.** Given a DEL dataset $\mathcal{D} = \{(\mathbf{f}_i, \mathbf{p}_i, M_i, R_i, y_i)\}_{i=1}^N$, where $\mathbf{f}_i \in \mathbb{R}^d$ denotes the molecular fingerprint, $\mathbf{p}_i \in \mathbb{R}^m$ represents the binding pose, $M_i \in \mathbb{R}$ is the matrix count derived from control experiments without protein targets, $R_i \in \mathbb{R}$ is the target count obtained from experiments involving protein target binding, and $y_i \in \{0, 1\}$ indicates the **functional group label**. We propose a joint multi-task learning framework $\mathcal{F} : \mathbb{R}^d \times \mathbb{R}^m \to \mathbb{R} \times \mathbb{R} \times [0, 1]$ such that:

$$\mathcal{F}(\mathbf{f}_i, \mathbf{p}_i) = (\hat{M}_i, \hat{R}_i, \hat{p}_i) \tag{1}$$

where $\hat{M}_i$ and $\hat{R}_i$ represent the predicted matrix count and target count; and $\hat{p}_i$ is the predicted likelihood of the functional group label. The primary focus of this framework lies in predicting accurate read count values that strongly correlate with the actual binding affinity ($K_i$ values).

**Zero-Inflated Poisson Distribution (ZIP) & ZIP Loss.** Zero-Inflated Poisson Distribution was applied to address **Distribution Noise** (Shmilovich et al., 2023; Lim et al., 2022), effectively modeling read counts $M_i$ and $R_i$ as Poisson distributions characterized by an excess frequency of zeros. By defining $r_i \in \{M_i, R_i\}$ and $\hat{r}_i \in \{\hat{M}_i, \hat{R}_i\}$ as the ground-truth and model's predicted read count values, we can express ZIP as:

$$P(X = r_i | \lambda, \pi) = \begin{cases} \pi + (1 - \pi)e^{-\lambda}, & \text{if } r_i = 0 \\ (1 - \pi)\frac{\lambda^{r_i} e^{-\lambda}}{r_i!}, & \text{if } r_i > 0 \end{cases} \tag{2}$$

where $\pi$ denotes the probability of excess zeros, and $\lambda$ denotes the mean parameter of the Poisson component. In (Shmilovich et al., 2023), the ZIP distributions of $M_i$ and $R_i$ are modeled using different $\pi$ values ($\pi_M$ and $\pi_R$), based on their respective orders of magnitude. The regression is achieved by minimizing the Negative Log-Likelihood (NLL) for all predicted read counts $\hat{M}_i$ and $\hat{R}_j$:

$$\mathcal{L}_{\text{ZIP}} = -\sum_i \log[P(\hat{M}_i | \lambda_M, \pi_M)] - \sum_j \log[P(\hat{R}_j | \lambda_M + \lambda_R, \pi_R)] \tag{3}$$

where $\lambda_M$ and $\lambda_R$ represent Poisson mean parameters for matrix and target counts, and $\pi_M$ and $\pi_R$ denote their respective zero-excess probabilities. This joint regression of target and control counts enable the model to learn the differential behavior of ligands in the presence and absence of targets, thereby potentially capturing the binding affinity.

$K_i$ **Estimation.** DEL read count prediction aims to estimate compound-target binding affinities ($K_i$ values) for drug candidate identification. We assess performance using Spearman rank correlation ($\rho_s$) between predicted read counts and experimental $K_i$ values: $\rho_s = 1 - \frac{6\sum_{i=1}^{n} d_i^2}{n(n^2-1)}$, where $n$ is the sample number and $d_i$ is the difference between the ranks of corresponding values in the two variables. Ideally, $K_i$ values and read counts are negatively correlated, as lower $K_i$ values indicate stronger binding affinity, which should correspond to higher read counts.

**LambdaRank** (Burges et al., 2006) Consider a ranking list with items $i = 1, \ldots, N$, relevance labels $y_i \in \mathbb{R}$ (larger is better), and model scores $s_i \in \mathbb{R}$. Sorting scores in descending order gives a permutation $\pi_s$ with $s_{\pi_s(1)} \geq \cdots \geq s_{\pi_s(N)}$; let $\Omega(i)$ be the rank position of item $i$ in $\pi_s$.

**Discounted cumulative gain.** Given a gain function $G : \mathbb{R} \to \mathbb{R}_{\geq 0}$, the discounted cumulative gain (DCG) of a permutation $\pi$ is

$$\text{DCG}(\pi; y) = \sum_{k=1}^{N} \frac{G(y_{\pi(k)})}{\log_2(1 + k)}. \tag{4}$$

The "ideal" permutation $\pi^\star$ is obtained by sorting items in non-increasing order of $y_i$, and its DCG,

$$\text{IDCG}(y) = \text{DCG}(\pi^\star; y), \tag{5}$$

is the maximum achievable DCG for the given labels. The normalized DCG (NDCG) of scores $s$ is then defined as

$$\text{NDCG}(s, y) = \frac{\text{DCG}(\pi_s; y)}{\text{IDCG}(y)} \in [0, 1], \tag{6}$$

where $\pi_s$ is the permutation induced by $s$. For items $i$ and $j$, $\Delta\text{NDCG}_{ij}(s, y)$ denotes the change in NDCG when swapping their positions in $\pi_s$.

**Pairwise objective.** For each pair $(i, j)$ with $y_i > y_j$, LambdaRank uses the RankNet logistic model with pairwise probability

$$P_{ij}(s) = \sigma(s_i - s_j) = \frac{1}{1 + \exp(-(s_i - s_j))}, \tag{7}$$

and pairwise loss

$$\ell_{ij}(s) = -\log P_{ij}(s) = \log(1 + \exp(-(s_i - s_j))). \tag{8}$$

The corresponding lambda-gradients are obtained by rescaling the RankNet gradients with the absolute NDCG change:

$$\lambda_{ij} = \left|\Delta\text{NDCG}_{ij}(s,y)\right| \frac{\partial \ell_{ij}}{\partial s_i}, \qquad \lambda_{ji} = -\lambda_{ij}, \tag{9}$$

and the total gradient for item $i$ is $\lambda_i = \sum_j \lambda_{ij}$. By construction, there exists an implicit loss whose gradient with respect to $s_i$ is $\lambda_i$, and minimizing this loss directly promotes improvements in NDCG.

**ListMLE** (Xia et al., 2008) Let $\pi^\star$ be the ground-truth permutation induced by $y_i$ (items sorted in non-increasing $y_i$). The Plackett–Luce probability of a permutation $\pi$ under scores $s$ is

$$P(\pi \mid s) = \prod_{k=1}^{N} \frac{\exp\big(s_{\pi(k)}\big)}{\sum_{j=k}^{N} \exp\big(s_{\pi(j)}\big)}. \tag{10}$$

ListMLE minimizes the negative log-likelihood of $\pi^\star$:

$$L_{\text{ListMLE}}(s; \pi^\star) = -\log P(\pi^\star \mid s) = -\sum_{k=1}^{N}\left[ s_{\pi^\star(k)} - \log\sum_{j=k}^{N} \exp\big(s_{\pi^\star(j)}\big)\right]. \tag{11}$$

In the next two subsections, we adapt these classical pairwise (LambdaRank) and listwise (ListMLE) formulations to DEL read-count data by taking model outputs $\hat{r}_i$ as scores $s_i$ and introducing DEL-specific weights and normalizations.

## 3.2 RANKING-BASED DISTRIBUTION NOISE REMOVAL

To effectively mitigate **Distribution Noise** in DEL read count data, we propose a novel ranking-based loss function $\mathcal{L}_{\text{rank}}$. This loss function integrates both local and global read count perspectives to fit the rank ordering of count values, resulting in a well-ordered ZIP that effectively captures the underlying read count pattern.

$$\mathcal{L}_{\text{rank}} = \beta\mathcal{L}_{\text{PSR}} + (1-\beta)\mathcal{L}_{\text{LGR}} \tag{12}$$

where $\beta \in [0,1]$ is a balancing hyperparameter that controls the relative contribution of the two components. $\mathcal{L}_{\text{PSR}}$ (PSR Loss) addresses local pairwise comparisons between compounds, while $\mathcal{L}_{\text{LGR}}$ (LGR Loss) captures global ranking information across the entire dataset. Together, they facilitate a well-ordered ZIP distribution for read count values. To establish the effectiveness of our ranking-based approach, we provide the following theoretical justification:

**Lemma 3.1.** *Given a set of feature-read count pairs $\{(x_i, r_i)\}_{i=1}^n$, where $x_i$ is the fused representation of sample $i$ derived from molecular fingerprint $f_i$ and binding pose $p_i$, and a well-fitted ZIP model $f_{ZIP}(r|x)$, the ranking loss $\mathcal{L}_{rank}$ provides positive information gain over the zero-inflated loss $\mathcal{L}_{ZIP}$:*

$$I(\mathcal{L}_{rank}|\mathcal{L}_{ZIP}) = H(R|\mathcal{L}_{ZIP}) - H(R|\mathcal{L}_{ZIP}, \mathcal{L}_{rank}) > 0$$

*where $H(R|\cdot)$ denotes the conditional entropy of read counts $R$.*

Building upon this information gain, we can further demonstrate that our combined approach, which incorporates both the zero-inflated and ranking losses, outperforming the standard zero-inflated model in terms of expected regression error. This enhancement is formalized in the following theorem:

**Theorem 3.2.** *Given a sufficiently large dataset $\{(x_i, r_i)\}_{i=1}^n$ of feature–read-count pairs, let $L_{ZIP}$ denote the loss function of the standard zero-inflated model and let $L_{rank}$ be a non–negative ranking loss. Let $\hat{r}^{ZIP}$ be the predictor that minimizes the expected ZIP loss and assume that the ranking loss is non–trivial at this predictor, i.e.*

$$\mathbb{E}\big[L_{rank}(\hat{r}^{ZIP}, R)\big] > 0.$$

*For any $\alpha \in (0,1)$, define the combined loss*

$$L_C(\hat{r}) = \alpha\, L_{ZIP}(\hat{r}) + (1-\alpha)\, L_{rank}(\hat{r}).$$

*Let $\hat{r}^C$ be a minimizer of the expected combined loss $\mathbb{E}[L_C(\hat{r})]$. Then there exists $\alpha^\star \in (0,1)$ such that*

$$\mathbb{E}\big[L_C(\hat{r}^C)\big] < \mathbb{E}\big[L_{ZIP}(\hat{r}^{ZIP})\big]. \tag{13}$$

*In particular, any constant predictor cannot minimize $L_C$, since its ranking loss is strictly positive.*

These theoretical results demonstrate that incorporating ranking information effectively aligns read counts across compounds, mitigating experimental biases in DEL screening data. The combined loss function consistently outperforms the standard ZIP approach in expected performance. Detailed proofs and analyses are provided in Sections A.1 and A.2.

### 3.2.1 PAIRWISE SOFT RANKING LOSS

To better model the relationships between compound pairs and handle read count noise, we introduce $\mathcal{L}_{\text{PSR}}$ inspired by LambdaRank (Burges et al., 2006). Compared to LambdaRank, it enables differentiable ranking between compounds, which is formulated as:

$$\mathcal{L}_{\text{PSR}}(\hat{r}_i, ..., \hat{r}_N, T) = -\sum_{i=1}^{N} \sum_{j \neq i, r_i > r_j} [\Delta_{ij} \cdot \sigma_{ij}(T)] \tag{14}$$

$$\sigma_{ij} = \frac{1}{1 + e^{-|r_i - r_j|/T}}, \quad \Delta_{ij} = \frac{\Delta G_{ij} \cdot \Delta D_{ij}}{Z}$$

where $\hat{r}_i$ and $\hat{r}_j$ represent the predicted read count values for compounds $i$ and $j$, respectively. $\sigma_{ij}$ reflects the absolute ranking differences between each compound pair, and $T$ denotes the temperature to scale the difference. For ranking changes, we introduce a pairwise importance term $\Delta_{ij}$ between compounds $i$ and $j$, comprising a gain function $G_i = \text{softplus}(r_i)$ for compound relevance and a rank-based discount function $D_i = 1/(\log_2(\text{rank}_i + 1) + \epsilon)$

$$\Delta G_{ij} = G_i - G_j = \text{softplus}(r_i) - \text{softplus}(r_j) \tag{15}$$

$$\Delta D_{ij} = D_i - D_j = \frac{1}{(\log_2(\text{rank}_i + 1) + \epsilon)} - \frac{1}{(\log_2(\text{rank}_j + 1) + \epsilon)} \tag{16}$$

, where $\epsilon$ ensures numerical stability; $\text{rank}_i$ denotes the predicted rank of sample i according to read-count values in each training batch. We then employ a normalization factor derived from the top-K predicted values per batch (K < N), improving computational efficiency and eliminating ranking noise from zero-value predictions.

$$Z = \sum_{k=1}^{K} \frac{\text{softplus}(\hat{r}_{[k]})}{\log_2(k + 1) + \epsilon} \tag{17}$$

where $\hat{r}_{[k]}$ represents the k-th highest predicted read count in descending order; $\epsilon$ is set to $1e - 8$ to avoid division by zero. This normalization factor adaptively adjusts the loss scale across different dataset sizes and read count distributions, ensuring robust model training regardless of data variations. $L_{PSR}$ extends LambdaRank by using continuous DEL read counts and Top-K–normalized NDCG weights, making the pairwise ranking robust to zero inflation and noisy count scales.

### 3.2.2 LISTWISE GLOBAL RANKING LOSS

We further propose $\mathcal{L}_{\text{LGR}}$ inspired by ListMLE (Xia et al., 2008) as a complement to $\mathcal{L}_{\text{PSR}}$. Compared to ListMLE, it is equipped with an additional loss term to distinguish excessive zero read-counts in DEL datasets. The LGR loss captures global ranking information as:

$$\mathcal{L}_{\text{LGR}}(\hat{r}, \tau, T) = -\sum_{i=1}^{N} \log \frac{\exp(\hat{r}_{\Omega(i)}/T)}{\sum_{j=i}^{N} \exp(\hat{r}_{\Omega(j)}/T)} + \sigma \sum_{i=1}^{N} \sum_{j>i} \mathcal{L}_{\text{con}}(\hat{r}_i, \hat{r}_j, \tau) \tag{18}$$

where $\Omega_i$ denotes the rank of compound i; $\tau$ represents the minimal margin between predicted read count pairs $(r_i, r_j)$; $T$ is a temperature parameter that rescales scores to sharpen the predicted read-count distribution; and $\mathcal{L}_{\text{con}}$ denotes a contrastive loss component that captures local relationships between ranking scores, weighted by parameter $\sigma$.

The contrastive loss function $\mathcal{L}_{\text{con}}$ is specifically designed to enhance discrimination between varying levels of biological activity, especially for samples with zero or identical read count values. Let $f : \mathbb{R} \to \mathbb{R}$ be the descending sorting function and $\tau > 0$ a fixed threshold. We define $\mathcal{L}_{\text{con}} : \mathbb{R} \times \mathbb{R} \times \mathbb{R}_{>0} \to \mathbb{R}_{\geq 0}$ as:

$$\mathcal{L}_{\text{con}}(\hat{r}_i, \hat{r}_j, \tau) = \max\{0, \tau - (f(\hat{r}_i) - f(\hat{r}_j))\} \tag{19}$$

This loss function is positive if and only if $f(\hat{r}_i) - f(\hat{r}_j) < \tau$, effectively enforcing a minimum margin $\tau$ between differently ranked samples. The constant gradients $\partial \mathcal{L}_{\text{con}} / \partial f(\hat{r}_i) = -1$ and $\partial \mathcal{L}_{\text{con}} / \partial f(\hat{r}_j) = 1$ when $f(\hat{r}_i) - f(\hat{r}_j) < \tau$ promote robust and stable ranking relationships, particularly beneficial for compounds with similar readouts but different underlying activities. $L_{LGR}$ extends ListMLE by adding a contrastive margin term that explicitly pushes high-count compounds above zero/near-zero ones, improving separation of truly active vs inactive DEL molecules.

### 3.3 CHEMICAL-REFERENCED DISTRIBUTION CORRECTION FRAMEWORK

To address **Distribution Shifts** in DEL, we propose the CRC framework, which enhances the read-count distribution by the functional group distribution alignment. We apply the Refinement-Correction optimization process (details in Algorithm1).

In the Refinement Stage, we apply dual information streams—chemical functional group labels and read counts—that respectively capture overall binding potential and binding strength. we adopt an iterative mechanism inspired by self-training techniques (Zoph et al., 2020) to update 2D SMILES embeddings and combined 2D-3D embeddings. Through multiple rounds of updates, the bidirectional feedback loop merge the information of the two representations.

In the Correction Stage, we introduce a consistency loss function to mitigate error accumulation and align predictions with underlying biological signals. Drawing on insights from (Hou et al., 2023), we leverage a key observation: specific functional groups within our dataset exhibit strong correlations with compound affinity, enabling us to define corresponding chemical group function labels. This labeling approach provides effective supervision for both read count regression and novel functional group discovery (Section 4.2). Importantly, this mechanism addresses discrepancies where compounds exhibit low read counts but high activity. Formally, the consistency loss is defined as:

$$\mathcal{L}_{\text{consist}}(r_i, \hat{r}_i, y_i, \hat{p}_i) = \|\hat{p}_i - y_i\| + \max\left(0, \|\hat{y}_i - \frac{\hat{r}_i}{\max_{i \in \{1,...,N\}} \hat{r}_i}\|_2^2 - \|y_i - \frac{r_i}{\max_{i \in \{1,...,N\}} r_i}\|_2^2\right)$$
(20)

where $N$ denotes batch size. The first term regresses functional group labels, while the second term constrains the consistency between normalized read counts and activity predictions. Unlike generic iterative self-training, CRC jointly refines the read-count and functional-group heads via a batch-wise consistency loss on fused 2D/3D embeddings, and Table 3, together with Appendix D.2, shows that disabling CRC or its refinement steps consistently reduces performance, especially under high noise.

### 3.4 TOTAL TRAINING OBJECTIVE

The total training objective integrates the ZIP loss, the ranking loss, and the consistency loss, which is formulated as:

$$\mathcal{L}_{\text{total}} = \mathcal{L}_{\text{ZIP}} + \rho \mathcal{L}_{\text{rank}} + \gamma \mathcal{L}_{\text{consist}}$$
(21)

where $\rho$ and $\gamma$ are hyperparameters that control the relative contribution of the ranking and consistency losses, respectively. These weights are tuned to balance the order of magnitude of each component for optimal performance across different experimental settings.

### 3.5 MODEL ARCHITECTURE

The DEL-Ranking framework consists of three components (Figure 2, Algorithm 3). The Fingerprint Encoder converts 2048-bit Morgan fingerprints to dense embeddings via residual MLP, capturing chemical substructure information. The Pose Encoder uses a pretrained 3D CNN from GNINA (McNutt et al., 2021) to encode protein-ligand complexes from 9-20 docked binding poses per compound. The Fusion Module combines pose and fingerprint information through self-attention, automatically prioritizing relevant poses without pose-level supervision. The model predicts matrix effects from fingerprint embeddings alone, while binding affinity and target effects use the fused representation. This design reflects that matrix binding depends primarily on molecular properties, whereas target binding requires specific 3D protein-ligand interactions. Implementation follows DEL-Dock (Shmilovich et al., 2023) architectural principles and hyperparameters.

Table 1: Comparison of our framework DEL-Ranking with existing DEL affinity predictions on CA2 & CA12 datasets. Results in **bold** and underlined are the top-1 and top-2 performances, respectively.

| Metric | 3p3h (CA2) | | 4kp5-A (CA12) | | 4kp5-OA (CA12) | | 5fl4-9p (CA9) | | 5fl4-20p (CA9) | |
|---|---|---|---|---|---|---|---|---|---|---|
| | Sp | SubSp | Sp | SubSp | Sp | SubSp | Sp | SubSp | Sp | SubSp |
| Mol Weight | -0.250 | -0.125 | -0.101 | 0.020 | -0.101 | 0.020 | -0.121 | -0.028 | -0.121 | -0.074 |
| Benzene | 0.022 | 0.072 | -0.054 | 0.035 | -0.054 | 0.035 | -0.174 | -0.134 | -0.199 | -0.063 |
| Smina Docking | $-0.174_{\pm0.002}$ | $-0.017_{\pm0.003}$ | $0.025_{\pm0.001}$ | $0.150_{\pm0.003}$ | $0.025_{\pm0.001}$ | $0.150_{\pm0.003}$ | $-0.114_{\pm0.009}$ | $-0.055_{\pm0.007}$ | $-0.279_{\pm0.044}$ | $-0.091_{\pm0.061}$ |
| RF-Enrichment | $-0.017_{\pm0.026}$ | $-0.042_{\pm0.025}$ | $-0.029_{\pm0.038}$ | $-0.005_{\pm0.048}$ | $\underline{-0.101_{\pm0.009}}$ | $-0.087_{\pm0.010}$ | $-0.064_{\pm0.126}$ | $-0.144_{\pm0.024}$ | $-0.064_{\pm0.126}$ | $-0.144_{\pm0.024}$ |
| RF-ZIP | $0.027_{\pm0.139}$ | $-0.005_{\pm0.071}$ | $0.035_{\pm0.094}$ | $-0.026_{\pm0.111}$ | $0.006_{\pm0.095}$ | $-0.021_{\pm0.122}$ | $0.040_{\pm0.022}$ | $-0.011_{\pm0.042}$ | $0.054_{\pm0.094}$ | $0.026_{\pm0.111}$ |
| MLP-ZIP | $-0.095_{\pm0.051}$ | $-0.085_{\pm0.115}$ | $-0.072_{\pm0.054}$ | $-0.058_{\pm0.093}$ | $-0.003_{\pm0.079}$ | $0.020_{\pm0.033}$ | $-0.029_{\pm0.094}$ | $-0.085_{\pm0.116}$ | $-0.055_{\pm0.066}$ | $-0.049_{\pm0.102}$ |
| Dos-DEL | $-0.048_{\pm0.036}$ | $-0.011_{\pm0.035}$ | $-0.016_{\pm0.029}$ | $-0.017_{\pm0.021}$ | $-0.003_{\pm0.030}$ | $-0.048_{\pm0.034}$ | $-0.115_{\pm0.065}$ | $-0.036_{\pm0.010}$ | $-0.231_{\pm0.007}$ | $-0.091_{\pm0.012}$ |
| DEL-QSVR | $-0.228_{\pm0.021}$ | $\underline{-0.171_{\pm0.033}}$ | $-0.004_{\pm0.178}$ | $0.018_{\pm0.139}$ | $0.070_{\pm0.134}$ | $-0.076_{\pm0.116}$ | $-0.086_{\pm0.060}$ | $-0.036_{\pm0.074}$ | $-0.298_{\pm0.005}$ | $-0.075_{\pm0.011}$ |
| DEL-Dock | $\underline{-0.255_{\pm0.009}}$ | $-0.137_{\pm0.012}$ | $\underline{-0.242_{\pm0.011}}$ | $\underline{-0.263_{\pm0.012}}$ | $0.015_{\pm0.029}$ | $\underline{-0.105_{\pm0.034}}$ | $\underline{-0.308_{\pm0.000}}$ | $\underline{-0.169_{\pm0.000}}$ | $\underline{-0.320_{\pm0.009}}$ | $\underline{-0.166_{\pm0.017}}$ |
| DEL-Ranking | $\mathbf{-0.286_{\pm0.002}}$ | $\mathbf{-0.177_{\pm0.005}}$ | $\mathbf{-0.268_{\pm0.012}}$ | $\mathbf{-0.277_{\pm0.016}}$ | $\mathbf{-0.289_{\pm0.025}}$ | $\mathbf{-0.233_{\pm0.021}}$ | $\mathbf{-0.323_{\pm0.015}}$ | $\mathbf{-0.175_{\pm0.000}}$ | $\mathbf{-0.330_{\pm0.007}}$ | $\mathbf{-0.187_{\pm0.013}}$ |

## 4 EXPERIMENT

**Datasets.** **CA9 Dataset** From the original data containing 108,529 DNA-barcoded molecules targeting human carbonic anhydrase IX (CA9) (Gerry et al., 2019), we derived two separate datasets. The first, denoted as **5fl4-9p**, uses 9 docked poses that we generated ourselves. The second, **5fl4-20p**, employs 20 docked poses using the 5fl4 structure. Both datasets lack chemical functional group labels. **CA2 and CA12 Datasets** From the CAS-DEL library (Hou et al., 2023), we generated three datasets comprising 78,390 molecules selected from 7,721,415 3-cycle peptide compounds. We performed docking to create 9 poses per molecule for each dataset. The CA2-derived dataset uses the 3p3h PDB structure (denoted as **3p3h**), while two CA12-derived datasets use the 4kp5 PDB structure: **4kp5-A** for normal expression and **4kp5-OA** for overexpression. The binary chemical functional group label is set to 1 when there is benzene sulfonamide (BB3-197) in the compound (Hou et al., 2023). **Validation Dataset** from ChEMBL (Zdrazil et al., 2024) includes 12,409 small molecules with affinity measurements for CA9, CA2, and CA12. Molecules have compatible atom types, molecular weights from 25 to 1000 amu, and inhibitory constants ($K_i$) from 90 pM to 0.15 M. A subset focusing on the 10-90th percentile range of the training data's molecular weights provides a more challenging test scenario. **Virtual Docking** details for ligand poses are shown in Appendix C.1.

**Evaluation Metrics and hyper-parameters.** We evaluate our framework on the ChEMBL dataset (Zdrazil et al., 2024) using two Spearman correlations: overall correlation ($\rho_{\text{overall}}$) between predicted read counts and experimental $K_i$ values across all validation data, and subset correlation ($\rho_{\text{subset}}$) for compounds with molecular weights between 10th-90th percentiles of training data. Detailed hyper-parameter settings are provided in Appendix C.2.

**Baselines.** We examine the performance of existing binding affinity predictors. Traditional methods based on binding poses and fingerprints include Molecule Weight, Benzene Sulfonamide, Smina Docking (Koes et al., 2013), and Dos-DEL(Gerry et al., 2019). AI-aided methods dependent on read count values and molecule information include RF-Enrichment, RF-ZIP (Random Forest for Log-enrichment, $\mathcal{L}_{\text{ZIP}}$), DEL-QSVR, and DEL-Dock (Lim et al., 2022; Shmilovich et al., 2023).

### 4.1 BENCHMARKING DENOISING CAPABILITY

**Benchmark Comparison.** We conducted comprehensive experiments across five diverse datasets: 3p3h, 4kp5-A, 4kp5-OA, and two variants of 5fl4. For each dataset, we performed five runs to ensure statistical robustness. As shown in Table 1, our method consistently achieves state-of-the-art results in both Spearman (**Sp**) and subset Spearman (**SubSp**) coefficients across all datasets.

Our analysis reveals several key insights: (1) **Experimental Adaptability**: DEL-Ranking shows consistent advantages across diverse datasets, with notable gains in challenging conditions. It maintains improvements even in lower-noise environments like purified protein datasets (3p3h and 5fl4), versatility highlighting DEL-Ranking's adaptability to various experimental setups. (2) **Noise Resilience**: DEL-Ranking excels in high-noise scenarios, particularly in membrane protein experiments. Its exceptional results on the 4kp5 dataset, especially the challenging 4kp5-OA variant, demonstrate this. Where baseline methods struggle, our approach effectively distinguishes signal from noise in complex experimental conditions. (3) **Structural Flexibility**: Our approach effectively uses structural information, as shown in the 5fl4 dataset. Increasing poses from 9 to 20 improves model

Table 2: Zero-shot Generalization Results Comparison evaluated on 3p3h, 4kp5-A, and 4kp5-OA datasets.

| | 3p3h (CA2) | | 4kp5-A (CA12) | | 4kp5-OA (CA12) | |
|---|---|---|---|---|---|---|
| Metric | Sp | SubSp | Sp | SubSp | Sp | SubSp |
| Mol Weight | -0.121 | -0.028 | -0.121 | -0.028 | -0.121 | -0.028 |
| Benzene | -0.174 | -0.134 | -0.174 | -0.134 | -0.174 | -0.134 |
| Smina Docking | $-0.114_{\pm 0.009}$ | $-0.055_{\pm 0.007}$ | $-0.114_{\pm 0.009}$ | $-0.055_{\pm 0.007}$ | $-0.114_{\pm 0.009}$ | $-0.055_{\pm 0.007}$ |
| RF-Enrichment | $0.020_{\pm 0.014}$ | $-0.031_{\pm 0.057}$ | $-0.034_{\pm 0.013}$ | $-0.034_{\pm 0.029}$ | $-0.044_{\pm 0.005}$ | $-0.085_{\pm 0.006}$ |
| RF-ZIP | $0.037_{\pm 0.059}$ | $0.013_{\pm 0.017}$ | $0.036_{\pm 0.024}$ | $-0.002_{\pm 0.016}$ | $0.049_{\pm 0.012}$ | $-0.007_{\pm 0.013}$ |
| Dos-DEL | $-0.115_{\pm 0.065}$ | $-0.036_{\pm 0.010}$ | $-0.115_{\pm 0.065}$ | $-0.036_{\pm 0.010}$ | $-0.115_{\pm 0.065}$ | $-0.036_{\pm 0.010}$ |
| DEL-QSVR | $-0.300_{\pm 0.020}$ | $\mathbf{-0.257_{\pm 0.022}}$ | $\mathbf{-0.236_{\pm 0.038}}$ | $\mathbf{-0.223_{\pm 0.030}}$ | $0.108_{\pm 0.089}$ | $0.130_{\pm 0.070}$ |
| DEL-Dock | $-0.272_{\pm 0.013}$ | $-0.118_{\pm 0.005}$ | $-0.211_{\pm 0.007}$ | $-0.118_{\pm 0.010}$ | $0.065_{\pm 0.021}$ | $-0.125_{\pm 0.034}$ |
| **DEL-Ranking** | $\mathbf{-0.310_{\pm 0.005}}$ | $-0.120_{\pm 0.011}$ | $-0.228_{\pm 0.010}$ | $-0.127_{\pm 0.018}$ | $\mathbf{-0.300_{\pm 0.026}}$ | $\mathbf{-0.129_{\pm 0.021}}$ |

performance, highlighting our method's ability to utilize additional structural data. This underscores DEL-Ranking's effectiveness in extracting insights from comprehensive structural information. (4) **Dual Analysis Capability**: DEL-Ranking's consistent performance in both Sp and SubSp metrics shows its versatility in drug discovery. This enables effective broad-spectrum screening and detailed subset analysis, enhancing its utility across various stages of drug discovery.

**Zero-shot Generalization.** We evaluated models' zero-shot generalization on CA9 by training them on CA2 and CA12 targets across three datasets (3p3h, 4kp5-A, and 4kp5-OA). Detailed in Table 2, DEL-Ranking consistently outperformed DEL-Dock. Notably, on the 4kp5-OA dataset with substantially different protein targets, DEL-Ranking maintained strong predictive performance, demonstrating its generalization capability to novel targets. Notably, DEL-QSVR exhibited superior zero-shot performance, suggesting that simpler molecular representations and loss functions might be more conducive to target generalization. This superior performance might be attributed to the fact that incorporating pose information could potentially limit zero-shot generalization capability.

Table 3: Ablation Study Results of DEL-Ranking on 3p3h, 4kp5-A, and 4kp5-OA datasets.

| | 3p3h (CA2) | | 4kp5-A (CA12) | | 4kp5-OA (CA12) | |
|---|---|---|---|---|---|---|
| Metric | Sp | SubSp | Sp | SubSp | Sp | SubSp |
| w/o All | $-0.255_{\pm 0.004}$ | $-0.137_{\pm 0.012}$ | $-0.242_{\pm 0.011}$ | $-0.263_{\pm 0.012}$ | $0.015_{\pm 0.029}$ | $-0.105_{\pm 0.034}$ |
| w/o $\mathcal{L}_{PSR}$ | $-0.273_{\pm 0.012}$ | $-0.155_{\pm 0.013}$ | $-0.251_{\pm 0.015}$ | $-0.271_{\pm 0.011}$ | $0.015_{\pm 0.028}$ | $-0.105_{\pm 0.033}$ |
| w/o $\mathcal{L}_{LGR}$ | $-0.280_{\pm 0.011}$ | $-0.168_{\pm 0.015}$ | $-0.256_{\pm 0.023}$ | $-0.273_{\pm 0.016}$ | $-0.269_{\pm 0.024}$ | $-0.209_{\pm 0.034}$ |
| w/o $\mathcal{L}_{con}$ | $-0.283_{\pm 0.004}$ | $-0.172_{\pm 0.007}$ | $-0.260_{\pm 0.018}$ | $-0.273_{\pm 0.014}$ | $-0.273_{\pm 0.024}$ | $-0.218_{\pm 0.034}$ |
| w/o Temp | $-0.279_{\pm 0.011}$ | $-0.166_{\pm 0.015}$ | $-0.247_{\pm 0.022}$ | $-0.265_{\pm 0.014}$ | $-0.256_{\pm 0.033}$ | $-0.181_{\pm 0.046}$ |
| w/o CRC | $-0.284_{\pm 0.007}$ | $-0.174_{\pm 0.010}$ | $-0.260_{\pm 0.015}$ | $-0.272_{\pm 0.012}$ | $-0.269_{\pm 0.023}$ | $-0.223_{\pm 0.045}$ |
| **DEL-Ranking** | $-0.286_{\pm 0.002}$ | $-0.177_{\pm 0.005}$ | $-0.268_{\pm 0.012}$ | $-0.277_{\pm 0.016}$ | $-0.289_{\pm 0.025}$ | $-0.233_{\pm 0.021}$ |

## 4.2 Discovery of Potential High Affinity Functional Group

To evaluate DEL-Ranking's capability in identifying potent compounds, we conducted an in-depth analysis of the Top-50 compounds predicted by our model across five datasets. These compounds were selected based on their predicted target counts in decreasing order.

Our primary objective was to examine the Ki values of these selected compounds to demonstrate that DEL-Ranking effectively identifies ligands with high binding affinity. Additionally, we performed a comparative analysis with DEL-Dock to highlight our model's enhanced ability to discover promising ligand candidates. Notably, our analysis revealed certain functional groups associated with low Ki values that were previously unreported by researchers in the 3p3h, 4kp5-A, and 4kp5-OA datasets. This finding demonstrates that our functional label approach and Consistency Regression Correction (CRC) mechanism extend beyond simply injecting known chemical biases such as Benzene Sulfonamide. As detailed in Figure 3, the selected compounds consistently exhibited low $K_i$ values across

all datasets, confirming the model's effectiveness in prioritizing high-affinity compounds from large DEL libraries.

**Benzene Sulfonamide Accuracy.** DEL-Ranking shows expectational accuracy in detecting benzene sulfonamide, a key high-affinity group for carbonic anhydrase inhibitors (Hou et al., 2023). From Figure 3, the model achieved high detection rates on five datasets, demonstrating that our CRC framework effectively incorporates biological prior knowledge into model prediction. To further explore the potential high-affinity compounds, we conducted the same study of DEL-Dock (Shmilovich et al., 2023) in Appendix D.3.

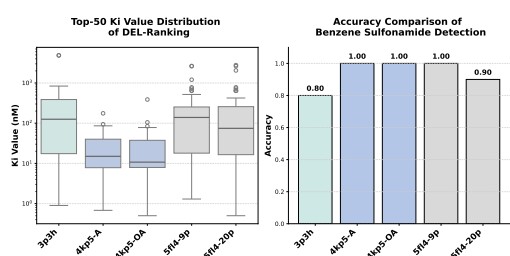

Figure 3: Quantitative analysis of Top-50 selection, including $K_i$ distribution and accuracy.

**Novel Group Discovery** Our analysis of the 3p3h and 5fl4 datasets revealed a significant finding: 20% (10/50) of high-ranking compounds in 3p3h and 10% (5/50) in 5fl4 lack the expected benzene sulfonamide group. Remarkably, all these compounds contain a common functional group - **Pyrimidine Sulfonamide** - which shares high structural similarity with benzene sulfonamide.

Further investigation through case-by-case $K_i$ value determination yielded compelling results. Five compounds from 3p3h and five from 5fl4 containing pyrimidine sulfonamide exhibited $K_i$ values comparable to or even surpassing those of benzene sulfonamide-containing compounds. This finding profoundly validates DEL-Ranking's dual capability: successfully incorporating chemical functional group label information, while simultaneously leveraging multi-level information along with integrated ranking orders to uncover potential high-activity functional groups. Notably, this discovery reveals DEL-Ranking's ability to identify unexplored scaffolds, showing potential to improve compound prioritization and accelerate hit-to-lead optimization in early-stage drug discovery. Detailed visualization of Top-50 samples and selected Pyrimidine Sulfonamide cases are shown in Sections D.5 and G. We anticipate that with increased sampling sizes, DEL-Ranking will demonstrate enhanced capability to identify additional affinity-determining functional groups that contribute significantly to binding interactions.

### 4.3 ABLATION STUDY

To further explore the effectiveness of our enhancement, we compare DEL-Ranking with some variants on 3p3h, 4kp5-A, and 4kp5-OA datasets. We can observe from Table 3 that (1) $\mathcal{L}_{\text{PSR}}$ and $\mathcal{L}_{\text{LGR}}$ contribute most significantly to model performance across all datasets. (2) The impact of $\mathcal{L}_{\text{PSR}}$ is more pronounced in datasets with higher noise levels, as evidenced by the larger relative performance drop in the 3p3h dataset. (3) Temperature adjustment and $\mathcal{L}_{\text{consist}}$ help improve the performance by correcting the predicted distributions, but count less than ranking-based denoising.

Furthermore, we conducted ablation studies on both loss weight and structure information weight (See details in Sections D.1 and D.2). Also, due to multiple hyperparameters, we provide hyperparameter selection criteria in AppendixC.2. The experimental results corroborate the capability of our approach and the feasibility of our hyperparameter selection criteria.

### 5 CONCLUSION

In this paper, we propose DEL-Ranking to address the challenge of noise in DEL screening through innovative ranking loss and activity-based correction algorithms. Experimental results demonstrate significant improvements in binding affinity prediction and generalization capability. Additionally, DEL-Ranking's ability to identify potential binding affinity determinants advances the field of DEL screening analysis by offering deeper insights into how molecular structures influence activity. Current limitations primarily stem from acquiring, integrating, and analyzing high-quality multi-modal molecular data at scale. Future work will focus on refining multi-modal data integration and expanding the model's interpretability to further advance DEL-based drug discovery and accelerate structure–activity relationship exploration.

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

# A   THEORETICAL ANALYSIS

## A.1   PROOF OF LEMMA AND THEOREM

*Proof.* **[Proof of Lemma 3.1]** Let $(\Omega, \mathcal{F}, P)$ be a probability space and $(X, R) : \Omega \to \mathcal{X} \times \mathbb{R}$ be random variables representing features and read counts respectively. Define $f_{\text{ZIP}}(r|x)$ as the probability mass function of a well-fitted Zero-Inflated Poisson model.

Define:

$$\hat{R}(x) = E[R|X = x] = \sum_{r=0}^{\infty} r \cdot f_{\text{ZIP}}(r|x)$$

$$\mathcal{L}_{\text{ZIP}}(f_{\text{ZIP}}, \mathcal{D}) = - \sum_{(x,r) \in \mathcal{D}} \log f_{\text{ZIP}}(r|x)$$

$$\mathcal{L}_{\text{rank}}(\hat{R}, \mathcal{D}) = \sum_{(x_i, r_i),(x_j, r_j) \in \mathcal{D}: r_i > r_j} \max(0, \hat{R}(x_j) - \hat{R}(x_i) + \delta)$$

where $\mathcal{D}$ is the observed dataset and $\delta > 0$.

We aim to prove $I(\mathcal{L}_{\text{rank}}|\mathcal{L}_{\text{ZIP}}) > 0$, where $I(\cdot|\cdot)$ denotes conditional mutual information.

Consider $(x_i, r_i), (x_j, r_j) \in \mathcal{D}$ with $r_i > r_j$. It's possible that $\hat{R}(x_i) \leq \hat{R}(x_j)$ due to the nature of likelihood optimization in the ZIP model. This occurs because ZIP optimization focuses on absolute likelihood values rather than preserving relative ordering between samples.

In such a case where observed ordering and predicted ordering disagree:

$$\mathcal{L}_{\text{ZIP}}(f_{\text{ZIP}}, \{(x_i, r_i), (x_j, r_j)\}) = - \log f_{\text{ZIP}}(r_i|x_i) - \log f_{\text{ZIP}}(r_j|x_j)$$

$$\mathcal{L}_{\text{rank}}(\hat{R}, \{(x_i, r_i), (x_j, r_j)\}) = \max(0, \hat{R}(x_j) - \hat{R}(x_i) + \delta) > 0$$

The ranking loss is positive specifically when the predicted ordering contradicts the observed ordering. During optimization, minimizing $\mathcal{L}_{\text{rank}}$ will push the model to correct such inversions, ensuring $\hat{R}(x_i) > \hat{R}(x_j)$ when $r_i > r_j$. This directly improves the model's ability to preserve ordering relationships.

Therefore, when both losses are used together:

$$P(R_i > R_j|\mathcal{L}_{\text{ZIP}}, \mathcal{L}_{\text{rank}}) > P(R_i > R_j|\mathcal{L}_{\text{ZIP}})$$

This inequality occurs because $\mathcal{L}_{\text{rank}}$ specifically penalizes ordering violations that $\mathcal{L}_{\text{ZIP}}$ alone might permit. Consequently, the conditional entropy decreases:

$$H(R|\mathcal{L}_{\text{ZIP}}, \mathcal{L}_{\text{rank}}) < H(R|\mathcal{L}_{\text{ZIP}})$$

Therefore, $I(\mathcal{L}_{\text{rank}}|\mathcal{L}_{\text{ZIP}}) = H(R|\mathcal{L}_{\text{ZIP}}) - H(R|\mathcal{L}_{\text{ZIP}}, \mathcal{L}_{\text{rank}}) > 0$. □

*Proof.* **[Proof of Theorem 3.2]** Given Lemma 3.1, we first prove that there exists a set of predictions $\hat{r}^C$ and a sufficiently small $\gamma_0 > 0$ such that for all $\gamma \in (0, \gamma_0)$:

$$E[\mathcal{L}_{\text{ZIP}}(\hat{r}^C, R)] - E[\mathcal{L}_{\text{ZIP}}(\hat{r}^{ZI}, R)] < \frac{1 - \gamma}{\gamma}(E[\mathcal{L}_{\text{rank}}(\hat{r}^{ZI}, R)] - E[\mathcal{L}_{\text{rank}}(\hat{r}^C, R)])$$

Define the combined loss function $L_C(\hat{r}, R; \alpha) = \alpha \mathcal{L}_{\text{ZIP}}(\hat{r}, R) + (1-\alpha)\mathcal{L}_{\text{rank}}(\hat{r}, R)$, where $\alpha \in (0, 1)$. Let $\hat{r}^C(\alpha)$ be the minimizer of $L_C$:

$$\hat{r}^C(\alpha) = \arg\min_{\hat{r}} E[L_C(\hat{r}, R; \alpha)]$$

By the definition of $\hat{r}^C(\alpha)$, for any $\alpha \in (0, 1)$, we have:

$$E[L_C(\hat{r}^C(\alpha), R; \alpha)] \leq E[L_C(\hat{r}^{\text{ZIP}}, R; \alpha)]$$

Expanding this inequality:

$$\alpha E[\mathcal{L}_{\text{ZIP}}(\hat{r}^C(\alpha), R)] + (1-\alpha)E[\mathcal{L}_{\text{rank}}(\hat{r}^C(\alpha), R)] \le \alpha E[\mathcal{L}_{\text{ZIP}}(\hat{r}^{\text{ZIP}}, R)] + (1-\alpha)E[\mathcal{L}_{\text{rank}}(\hat{r}^{\text{ZIP}}, R)]$$

Let $\Delta\mathcal{L}_{\text{ZIP}}(\alpha) = E[\mathcal{L}_{\text{ZIP}}(\hat{r}^C(\alpha), R)] - E[\mathcal{L}_{\text{ZIP}}(\hat{r}^{\text{ZIP}}, R)]$ and $\Delta\mathcal{L}_{\text{rank}}(\alpha) = E[\mathcal{L}_{\text{rank}}(\hat{r}^{\text{ZIP}}, R)] - E[\mathcal{L}_{\text{rank}}(\hat{r}^C(\alpha), R)]$. Rearranging the inequality:

$$\alpha\Delta\mathcal{L}_{\text{ZIP}}(\alpha) \le (1-\alpha)\Delta\mathcal{L}_{\text{rank}}(\alpha)$$

From Lemma 3.1, we established that $I(\mathcal{L}_{\text{rank}}|\mathcal{L}_{\text{ZIP}}) > 0$, meaning $\mathcal{L}_{\text{rank}}$ provides information not captured by $\mathcal{L}_{\text{ZIP}}$. This additional information allows the combined model to better preserve ranking relationships. Consequently, there exists $\alpha_1 \in (0, 1)$ such that for all $\alpha \in (0, \alpha_1]$, $\Delta\mathcal{L}_{\text{rank}}(\alpha) > 0$.

This positive $\Delta\mathcal{L}_{\text{rank}}(\alpha)$ occurs because the combined model $\hat{r}^C(\alpha)$ incorporates ordering constraints that directly improve ranking performance compared to the ZIP-only model $\hat{r}^{\text{ZIP}}$.

Now, consider the function:

$$f(\alpha) = (1-\alpha)\Delta\mathcal{L}_{\text{rank}}(\alpha) - \alpha\Delta\mathcal{L}_{\text{ZIP}}(\alpha)$$

We know that $f(\alpha) \ge 0$ for all $\alpha \in (0, 1)$ from the earlier inequality. Moreover, $f(0) = \Delta\mathcal{L}_{\text{rank}}(0) > 0$. This inequality holds at $\alpha = 0$ because the pure ranking model optimizes solely for order preservation, significantly outperforming the ZIP model in terms of ranking metrics.

By the continuity of $f(\alpha)$ and since $f(0) > 0$, there exists $\alpha_0 \in (0, \alpha_1]$ such that for all $\alpha \in (0, \alpha_0]$:

$$f(\alpha) > 0$$

This implies:

$$(1-\alpha)\Delta\mathcal{L}_{\text{rank}}(\alpha) > \alpha\Delta\mathcal{L}_{\text{ZIP}}(\alpha)$$

Dividing both sides by $\alpha(1-\alpha)$ (which is positive for $\alpha \in (0, 1)$):

$$\frac{\Delta\mathcal{L}_{\text{rank}}(\alpha)}{\alpha} > \frac{\Delta\mathcal{L}_{\text{ZIP}}(\alpha)}{1-\alpha}$$

This is equivalent to:

$$\Delta\mathcal{L}_{\text{ZIP}}(\alpha) < \frac{1-\alpha}{\alpha}\Delta\mathcal{L}_{\text{rank}}(\alpha)$$

Substituting back the definitions of $\Delta\mathcal{L}_{\text{ZIP}}(\alpha)$ and $\Delta\mathcal{L}_{\text{rank}}(\alpha)$:

$$E[\mathcal{L}_{\text{ZIP}}(\hat{r}^C(\alpha), R)] - E[\mathcal{L}_{\text{ZIP}}(\hat{r}^{\text{ZIP}}, R)] < \frac{1-\alpha}{\alpha}(E[\mathcal{L}_{\text{rank}}(\hat{r}^{\text{ZIP}}, R)] - E[\mathcal{L}_{\text{rank}}(\hat{r}^C(\alpha), R)])$$

Let $\hat{r}^C = \hat{r}^C(\alpha_0)$, where $\alpha_0$ represents the optimal trade-off between ZIP fidelity and ranking performance. At this value, we have:

$$E[\mathcal{L}_{\text{ZIP}}(\hat{r}^C, R)] - E[\mathcal{L}_{\text{ZIP}}(\hat{r}^{\text{ZIP}}, R)] < \frac{1-\alpha}{\alpha}(E[\mathcal{L}_{\text{rank}}(\hat{r}^{\text{ZIP}}, R)] - E[\mathcal{L}_{\text{rank}}(\hat{r}^C, R)])$$

Rearranging this inequality:

$$\alpha E[\mathcal{L}_{\text{ZIP}}(\hat{r}^C, R)] + (1-\alpha)E[\mathcal{L}_{\text{rank}}(\hat{r}^C, R)] < \alpha E[\mathcal{L}_{\text{ZIP}}(\hat{r}^{\text{ZIP}}, R)] + (1-\alpha)E[\mathcal{L}_{\text{rank}}(\hat{r}^{\text{ZIP}}, R)]$$

The left-hand side of this inequality is $E[L_C(\hat{r}^C)]$ by definition. The right-hand side is strictly greater than $E[\mathcal{L}_{\text{ZIP}}(\hat{r}^{\text{ZIP}})]$ since $E[\mathcal{L}_{\text{rank}}(\hat{r}^{\text{ZIP}}, R)] > 0$ for any non-trivial ranking loss and $\alpha < 1$.

Therefore:

$$E[L_C(\hat{r}^C)] < \alpha E[\mathcal{L}_{\text{ZIP}}(\hat{r}^{\text{ZIP}}, R)] + (1-\alpha)E[\mathcal{L}_{\text{rank}}(\hat{r}^{\text{ZIP}}, R)] < E[\mathcal{L}_{\text{ZIP}}(\hat{r}^{\text{ZIP}})]$$

This completes the proof, showing that our combined model achieves lower expected loss than the standard ZIP model by effectively balancing distribution modeling and ranking preservation. $\qquad\square$

## A.2 GRADIENT ANALYSIS

We analyze the composite ranking loss function $\mathcal{L}_{\text{rank}}$, which combines Pairwise Soft Ranking Loss and Listwise Global Ranking Loss. The gradient of $\mathcal{L}_{\text{rank}}$ with respect to $\hat{r}_i$ is:

$$\frac{\partial \mathcal{L}_{\text{rank}}}{\partial \hat{r}_i} = \beta \frac{\partial \mathcal{L}_{\text{PSR}}}{\partial \hat{r}_i} + (1 - \beta) \frac{\partial \mathcal{L}_{\text{LGR}}}{\partial \hat{r}_i} \tag{22}$$

$$\frac{\partial \mathcal{L}_{\text{PSR}}}{\partial \hat{r}_i} = -\left( \sum_{j \neq i} (\Delta_{ij} \cdot \sigma_{ij}) - \sum_{j \neq i} (\Delta_{ji} \cdot \sigma_{ji}) \right) - \hat{r}_i \sum_{j \neq i} \Delta_{ij} \cdot \frac{\partial \sigma_{ij}}{\partial \hat{r}_i} + \hat{r}_i \sum_{j \neq i} \Delta_{ji} \cdot \frac{\partial \sigma_{ji}}{\partial \hat{r}_i} \tag{23}$$

where

$$\frac{\partial \sigma_{ij}}{\partial \hat{r}_i} = \frac{\text{sign}(\hat{r}_i - \hat{r}_j)}{T} \sigma_{ij}(1 - \sigma_{ij}) \tag{24}$$

The gradient $\frac{\partial \mathcal{L}_{\text{PSR}}}{\partial \hat{r}_i}$ is primarily determined by $\Delta_{ij}$ and $\sigma_{ij}$, which represent pairwise comparisons between item $i$ and other items $j$. $\Delta_{ij}$ captures the NDCG impact of swapping items $i$ and $j$, while $\sigma_{ij}$ adjusts this impact based on the difference between $\hat{r}_i$ and $\hat{r}_j$. This formulation ensures that $\mathcal{L}_{\text{PSR}}$ focuses on local ranking relationships, particularly between adjacent or nearby items.

$$\frac{\partial \mathcal{L}_{\text{LGR}}}{\partial \hat{r}_i} = -\frac{1}{T} \sum_{k=i}^{n} \left( \frac{\exp(\hat{r}_{\pi(k)}/T)}{\sum_{j=k}^{n} \exp(\hat{r}_{\pi(j)}/T)} - \mathbb{K}[\pi(k) = i] \right) + \frac{\partial \mathcal{L}_{\text{con}}}{\partial \hat{r}_i} \tag{25}$$

The gradient $\frac{\partial \mathcal{L}_{\text{LGR}}}{\partial \hat{r}_i}$ incorporates information from all items ranked from position $i$ to $n$. Through its softmax formulation, it considers the position of item $i$ relative to all items ranked below it. This allows $\mathcal{L}_{\text{LGR}}$ to capture global ranking information.

## B DETAILED ALGORITHM OF CRC

---

**Algorithm 1** Refinement Stage for Chemical-Referenced Correction (CRC) Algorithm

---

**Require:** Pose structure embeddings $\mathbf{h}_p$, Fingerprint sequence embeddings $\mathbf{h}_f$, Sequence-structure balancing weight $\varsigma$, num_iterations $\mathbf{n}$, use_feedback
1: $x \leftarrow \texttt{PostAddLayer}(\varsigma \mathbf{h}_p + \mathbf{h}_f)$
2: $\hat{M} \leftarrow \texttt{MatrixHead}(\mathbf{h}_f)$
3: Initialize $\hat{R} \leftarrow \mathbf{0}, \hat{y} \leftarrow \mathbf{0}$
4: **for** $i = 1$ to $\mathbf{n}$ **do**
5:     **if** use_feedback **then**
6:         $\hat{R} \leftarrow [x; \hat{p}], \hat{p} \leftarrow [x; \hat{R}]$
7:     **else**
8:         $\hat{R} \leftarrow x, \hat{p} \leftarrow x$
9:     **end if**
10:    $\hat{R} \leftarrow \texttt{EnrichmentHead}(\texttt{ReadHead}(\hat{R}))$
11:    $\hat{p} \leftarrow \texttt{ActHead}(\hat{p})$
12: **end for**
13: **Return** $\hat{M}, \hat{R}, \hat{p}$

---

`PostAddLayer`, `MatrixHead`, `EnrichmentHead`, `ReadHead`, and `ActHead` are Multi-layer Perceptrons (MLP) that map latent embeddings into corresponding predicted count values and activities. The architectures are consistent with DEL-Dock (Shmilovich et al., 2023).

## C EXPERIMENTAL SETTINGS

### C.1 VIRTUAL DOCKING FOR DATASET CONSTRUCTION

We employed molecular docking to define the three-dimensional conformations of molecules within our DEL datasets. This method was applied to both the training and evaluation sets, generating

ligand binding poses for all molecules. We concentrated on three pivotal carbonic anhydrase proteins: Q16790 (CAH9_HUMAN), P00918 (CAH2_HUMAN), and O43570 (CAH12_HUMAN).

For the Q16790 target, we sourced the 5fl4 and 2hkf PDB structures from the PDBbind database and utilized the Gerry dataset (Gerry et al., 2019). which comprised 108,529 molecules, generating up to nine potential poses per molecule. For the targets P00918 and O43570, we selected 127,500 SMILES strings from the DEL-MAP dataset (Hou et al., 2023) and conducted self-docking using the 3p3h and 5doh PDB structures for P00918, and 4kp5 and 4ht2 for O43570, as sourced from PDBbind. For the validation set, we applied the same docking methodology to the corresponding ligands of CA9, CA2, and CA12, involving 3,324, 6,395, and 2,690 ligands respectively.

In the specific docking procedures, initial 3D conformations of ligands were created using RDKit. The binding sites in the protein-ligand complexes were identified using 3D structural data of known binding ligands from PDBbind as reference points. Targeted docking was performed by defining the search space as a 22.5 Å cube centered on the reference ligand in the corresponding PDBbind complex. Using SMINA docking software, we generated 9 potential poses for each protein-ligand pair.

## C.2    Hyperparameter Setting

The model was trained using the Adam optimizer with mini-batches of 64 samples. The network architecture employed a hidden dimension of 128. The self-correction mechanism was applied for 3 iterations. All experiments were conducted on a single NVIDIA RTX 3090 GPU with 24GB memory. The implementation utilized PyTorch-Lightning version 1.9.0 to streamline the training process and enhance reproducibility. The hyperparameter settings for different datasets, including loss function weights, temperature, and margin, are detailed in Table C.2.

**Hyper-parameter Selection**    The hyperparameter configuration in the appendix requires clarification regarding the weight settings. The key parameters include $\mathcal{L}_{\text{rank}}$ weight, $\mathcal{L}_{\text{PSR}}$ weight, $\mathcal{L}_{\text{LGR}}$ weight, and CRC weight. The $\mathcal{L}_{\text{rank}}$ weight is logarithmically distributed between 1e9 and 1e11 to align with the magnitude of ZIP loss. $\mathcal{L}_{\text{PSR}}$ and $\mathcal{L}_{\text{LGR}}$ weights are calibrated to maintain appropriate balance among different ranking objectives. Given that CRC loss naturally aligns with ZIP loss magnitude, its weight is simply set to 1.0 or 0.1.

Temperature settings are determined by the characteristics of DEL read count data distribution, with denser distributions requiring lower temperatures. A detailed analysis of read count distribution and supporting theoretical proposition are provided in the Section D.1. Besides, the contract weight and margin serve as penalty terms for $\mathcal{L}_{\text{LGR}}$, with the weight selected based on $\mathcal{L}_{\text{LGR}}$'s relative magnitude. Detailed in Table C.2, these values remain stable and consistent across experiments.

Table 4: Hyperparameter Settings for DEL-Ranking on Different Datasets

|  | 3p3h | 4kp5-A | 4kp5-OA | 5fl4-9p | 5fl4-20p |
|---|---|---|---|---|---|
| $\mathcal{L}_{\text{consist}}$ weight $\gamma$ | 1 | 0.1 | 0.1 | – | – |
| $\mathcal{L}_{\text{rank}}$ weight $\rho$ | $1e11$ | $1e9$ | $1e10$ | $1e8$ | $1e8$ |
| $\mathcal{L}_{\text{PSR}}$ weight $\beta$ | 0.5 | 0.91 | 0.91 | 0.67 | 0.5 |
| $\mathcal{L}_{\text{LGR}}$ weight $1\text{-}\beta$ | 0.5 | 0.09 | 0.09 | 0.33 | 0.5 |
| Temperature $T$ | 0.8 | 0.3 | 0.2 | 0.9 | 0.2 |
| $\mathcal{L}_{\text{con}}$ weight $\sigma$ | $1e-3$ | $1e-3$ | $1e-3$ | $1e-4$ | $1e-3$ |
| Margin $\tau$ | $1e-3$ | $1e-3$ | $1e-3$ | $1e-3$ | $1e-3$ |

**Proposition C.1.** *As $T \to 0$, the model simultaneously achieves: (1)**Distributional Consistency** ensures high predicted read counts align with true binding affinities, identifying top-ranked compounds with the strongest binding potential; (2) **Increased robustness** mitigates the impact of small noise perturbations in experimental data.*

Based on the Proposition, the adaptive-ranking model would obtain more consistent identification of high-affinity compounds, reducing errors due to random fluctuations. Also, it achieves enhanced robustness against common DEL experimental noises such as PCR bias and sequencing errors. While

lowering the temperature leads to a more deterministic ranking with high-affinity sensitivity and noise resistance, there exists overlooking of compounds with slightly lower rankings when the temperature goes to extremely low. In experiments, we demonstrate that [0.1, 0.4] should be a proper range for the distribution sharping.

# D  EXPERIMENTAL RESULTS

Table 5: Comparison of different hyper-parameters on binding affinity prediction performance. The best performance within one set of hyperparameter group is set **bold**.

| Parameter | Value | 3p3h (CA2) | | 4kp5-A (CA12) | | 4kp5-OA (CA12) | |
|---|---|---|---|---|---|---|---|
| Metric | | Sp | SubSp | Sp | SubSp | Sp | SubSp |
| $\mathcal{L}_{\text{consist}}$ weight $\gamma$ | 0.1 | -0.275±$_{0.011}$ | -0.163±$_{0.017}$ | **-0.268±$_{0.012}$** | **-0.277±$_{0.016}$** | **-0.289±$_{0.025}$** | **-0.233±$_{0.021}$** |
| | 1 | **-0.286±$_{0.002}$** | **-0.177±$_{0.005}$** | -0.266±$_{0.008}$ | -0.238±$_{0.008}$ | -0.287±$_{0.005}$ | -0.213±$_{0.014}$ |
| | 10 | -0.276±$_{0.010}$ | -0.163±$_{0.015}$ | -0.258±$_{0.019}$ | -0.239±$_{0.010}$ | -0.278±$_{0.024}$ | -0.227±$_{0.040}$ |
| $\mathcal{L}_{\text{rank}}$ weight $\rho$ | 1e9 | -0.266±$_{0.011}$ | -0.151±$_{0.016}$ | **-0.268±$_{0.012}$** | **-0.277±$_{0.016}$** | -0.152±$_{0.045}$ | -0.225±$_{0.023}$ |
| | 1e10 | -0.269±$_{0.006}$ | -0.151±$_{0.009}$ | -0.257±$_{0.005}$ | -0.189±$_{0.016}$ | **-0.289±$_{0.025}$** | **-0.233±$_{0.021}$** |
| | 1e11 | **-0.286±$_{0.002}$** | **-0.177±$_{0.005}$** | -0.135±$_{0.012}$ | -0.060±$_{0.036}$ | -0.084±$_{0.095}$ | -0.058±$_{0.077}$ |
| $\mathcal{L}_{\text{LGR}}$ weight $\beta$ | 0.09 | -0.277±$_{0.009}$ | -0.165±$_{0.013}$ | **-0.268±$_{0.012}$** | **-0.277±$_{0.016}$** | **-0.289±$_{0.025}$** | **-0.233±$_{0.021}$** |
| | 0.5 | **-0.286±$_{0.002}$** | **-0.177±$_{0.005}$** | -0.267±$_{0.033}$ | -0.240±$_{0.016}$ | -0.288±$_{0.025}$ | -0.247±$_{0.019}$ |
| | 0.91 | -0.275±$_{0.011}$ | -0.160±$_{0.019}$ | -0.173±$_{0.054}$ | -0.089±$_{0.038}$ | -0.279±$_{0.007}$ | -0.222±$_{0.033}$ |
| Temperature $T$ | 0.2 | -0.280±$_{0.021}$ | -0.173±$_{0.029}$ | -0.267±$_{0.013}$ | -0.247±$_{0.009}$ | **-0.289±$_{0.025}$** | **-0.233±$_{0.021}$** |
| | 0.5 | -0.279±$_{0.009}$ | -0.169±$_{0.014}$ | -0.266±$_{0.014}$ | -0.236±$_{0.012}$ | -0.275±$_{0.013}$ | -0.216±$_{0.005}$ |
| | 0.8 | **-0.286±$_{0.002}$** | **-0.177±$_{0.005}$** | -0.268±$_{0.010}$ | -0.222±$_{0.010}$ | -0.275±$_{0.035}$ | -0.220±$_{0.029}$ |
| | 1 | -0.279±$_{0.011}$ | -0.166±$_{0.015}$ | -0.247±$_{0.022}$ | -0.265±$_{0.014}$ | -0.256±$_{0.033}$ | -0.181±$_{0.046}$ |

## D.1  ABLATION STUDY ON HYPER-PARAMETERS

In order to evaluate the robustness of our method, we conduct a comprehensive analysis of four critical hyperparameters: the consistency loss weight $\gamma$, ranking loss weight $\rho$, LGR loss weight $\beta$, and temperature $T$ across three datasets (3p3h, 4kp5-A, and 4kp5-OA). As shown in Table 5, we employ logarithmic search spaces for all loss-related hyperparameters to align the magnitudes of ranking and consistency losses with the ZIP loss, while adopting a linear search space for temperature.

The empirical results demonstrate that our selected hyperparameters consistently achieve optimal performance across all search spaces. The model exhibits strong stability, with performance variations remaining minimal under most hyperparameter adjustments. Nevertheless, we observe dataset-specific sensitivities: the 4kp5-OA dataset shows increased sensitivity to ranking loss weight variations, potentially due to elevated read count noise levels. Similarly, the 4kp5-A dataset exhibits performance fluctuations at higher values of ranking loss and $\mathcal{L}_{\text{LGR}}$ weights, which we attribute to magnitude imbalances in the numerical representations.

The performance progression with respect to temperature demonstrates a consistent linear relationship, providing empirical support for our distribution sharpening hypothesis. These findings collectively indicate that while our model maintains robustness across the hyperparameter search space with well-justified parameter selections, its sensitivity can be influenced by dataset-specific characteristics, particularly read count distribution noise and magnitude disparities in the underlying data.

## D.2  ABLATION STUDY ON STRUCTURE INFORMATION

To assess the value of structural information from docking software and its complementarity with sequence features, we performed an ablation study focusing on the additive combination of structure and fingerprint embeddings in the CRC algorithm. We applied varying scaling factors (0, 0.3, 0.6, 1.0, 1.5, and 2.0) to the structure embedding across three datasets (3p3h, 4kp5-A, and 4kp5-OA) with five random seeds. Table 6 shows that incorporating structural information significantly improves

Table 6: Parameter value comparison for structure scaling factor. The best performance within one set of hyperparameter group is set **bold**.

| Value $\varsigma$ | 3p3h (CA2) | | 4kp5-A (CA12) | | 4kp5-OA (CA12) | |
|---|---|---|---|---|---|---|
| Metric | Sp | SubSp | Sp | SubSp | Sp | SubSp |
| 0 | $-0.236\pm_{0.010}$ | $-0.112\pm_{0.013}$ | $-0.253\pm_{0.012}$ | $-0.218\pm_{0.017}$ | $-0.195\pm_{0.044}$ | $-0.103\pm_{0.055}$ |
| 0.3 | $-0.262\pm_{0.008}$ | $-0.145\pm_{0.012}$ | $-0.265\pm_{0.017}$ | $-0.227\pm_{0.017}$ | $-0.124\pm_{0.146}$ | $-0.062\pm_{0.090}$ |
| 0.6 | $-0.263\pm_{0.008}$ | $-0.146\pm_{0.011}$ | $-0.250\pm_{0.017}$ | $-0.231\pm_{0.019}$ | $-0.210\pm_{0.040}$ | $-0.121\pm_{0.047}$ |
| 1 | $\mathbf{-0.286\pm_{0.002}}$ | $\mathbf{-0.177\pm_{0.005}}$ | $\mathbf{-0.268\pm_{0.012}}$ | $\mathbf{-0.277\pm_{0.016}}$ | $\mathbf{-0.289\pm_{0.025}}$ | $\mathbf{-0.233\pm_{0.021}}$ |
| 1.5 | $-0.270\pm_{0.011}$ | $-0.155\pm_{0.016}$ | $-0.244\pm_{0.022}$ | $-0.252\pm_{0.022}$ | $-0.152\pm_{0.156}$ | $-0.139\pm_{0.104}$ |
| 2 | $-0.271\pm_{0.012}$ | $-0.155\pm_{0.015}$ | $-0.191\pm_{0.089}$ | $-0.216\pm_{0.060}$ | $-0.230\pm_{0.038}$ | $-0.152\pm_{0.051}$ |

model performance. The analysis revealed higher model sensitivity in the noise-prone 4kp5-OA dataset, while performance degradation was observed in 4kp5-A when scaling factors exceeded 1.0. These results indicate that while structural information enhances model performance, excessive weighting of potentially uncertain structural data can impair predictions. Nevertheless, our chosen parameterization demonstrates consistent performance across all datasets.

### D.3 COMPARISON RESULT OF TOP-50 SELECTION CASES BY DEL-DOCK

In evaluating the performance of DEL-Dock, a clear trend emerges across different datasets, driven largely by how each method—docking-based versus ranking-based—responds to varying noise levels and read counts. In the 3p3h dataset, DEL-Dock's ability to exploit direct protein–ligand interaction data helped it surpass DEL-Ranking in identifying compounds with low Ki values and benzenesulfonamide functionalities. In more moderate datasets, such as 4kp5-A and 5fl4-9p, both methods performed comparably, indicating that when the complexity and noise of the library remain at manageable levels, structure-based docking can achieve results on par with label-guided, ranking-focused

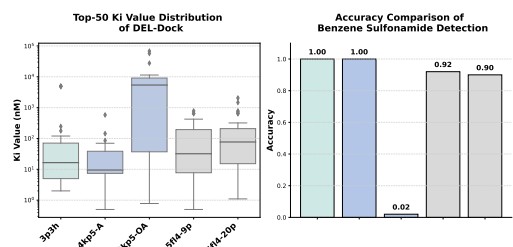

Figure 4: Quantitative analysis of Top-50 selection, including $K_i$ distribution and accuracy for DEL-Dock (Shmilovich et al., 2023).

algorithms. However, in the higher-noise 4kp5-OA and 5fl4-20p datasets, DEL-Ranking excelled—a finding consistent with theoretical expectations that label-driven methods, alongside increased read counts, are more robust to noisy environments.

### D.4 TRAINING TIME COMPARISON

We evaluate the computational efficiency by comparing the training time per epoch on a single NVIDIA RTX 3090 GPU. DEL-Dock and DEL-Ranking exhibit comparable time complexity; however, both incur higher computational costs than MLP-ZIP. This difference arises because MLP-ZIP bypasses the computationally intensive pose structure processing and fusion mechanisms.

Table 7: Comparison for training time per epoch.

| Method | Time (min) |
|---|---|
| MLP-ZIP | 1.88 |
| DEL-Dock | 2.31 |
| DEL-Ranking | 2.50 |

### D.5 VISUALIZATION OF TOP-50 SELECTION OF DEL-RANKING

Further reinforcing these observations is the discovery of thiocarbonyl and sulfonamide scaffolds with Ki values below 10.0, despite lacking the benzenesulfonamide functional group. DEL-Dock's success in identifying these structurally distinct, high-affinity compounds illustrates the capacity of docking-based approaches to uncover novel chemical scaffolds when protein–ligand interactions are well captured. Meanwhile, DEL-Ranking's aptitude for recognizing functional motifs analogous to benzenesulfonamides demonstrates how label guidance can extend to related binding groups. These combined insights point to a complementary dynamic between the two techniques: while direct docking may excel in less noisy settings or when the underlying structural biology is well-defined, ranking-based methods can leverage label data and higher read counts to maintain performance under more complex, noise-prone conditions. Below are the visualizations of the top-50 DEL-

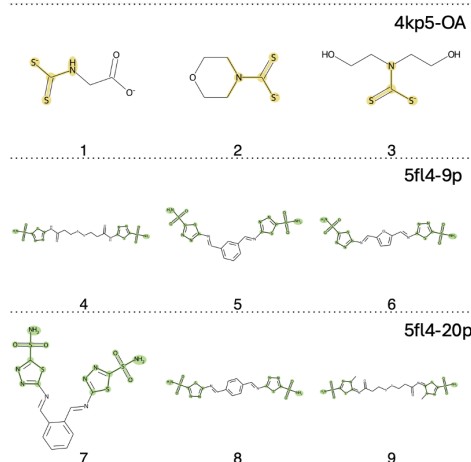

Figure 5: Visualization of Top-50 high affinity cases without benzene sulfonamide.

Ranking results across the 3p3h(Figure 6), 4kp5-A(Figure 7), 4kp5-OA(Figure 8), 5fl4(9 pose) in Figure 9, and 5fl4(20 pose) in Figure 10. In these figures, we specifically highlight benzenesulfonamide functional groups wherever they appear. While many of the top-ranked compounds do contain benzenesulfonamides—underscoring the importance of this moiety—there remain notable high-affinity hits devoid of benzenesulfonamide, suggesting that chemical diversity within the library can be harnessed to discover alternative active scaffolds in Figure 5 and Figure 4. By emphasizing the presence (or absence) of benzenesulfonamide in each molecule, these visualizations enable a clearer structural comparison across different binding poses, highlighting both the value of known functional groups and the potential for uncovering new ones.

# E    DISCUSSIONS

## E.1    LIMITATIONS

Despite DEL-Ranking's effectiveness in handling well-ordered read count distributions and its robustness against highly noisy datasets, several limitations merit consideration. First, the method necessitates prior knowledge of affinity-determining functional groups as correlation labels—information frequently unavailable in many DEL datasets. Second, DEL-Ranking's computational framework, which relies on protein-ligand binding poses and molecular fingerprinting, demands substantial computational resources, limiting its scalability to large-scale DEL libraries. In such cases, DEL-Ranking must be implemented as a two-stage training framework. Third, the magnitude of ranking loss varies considerably across different DEL datasets due to inherent differences in read count distributions, necessitating dataset-specific hyperparameter optimization.

## E.2    BOARDER IMPACTS

The DEL-Ranking framework has the potential to accelerate drug discovery by enabling more accurate identification of high-potency compounds. Additionally, its inherent ranking methodology facilitates the discovery of novel affinity-determining functional groups, thereby enhancing researchers' biological understanding of diverse protein-ligand systems. Furthermore, our curated DEL datasets contribute to the advancement of DEL denoising methodologies. Nevertheless, this framework presents minimal risk of misuse for developing compounds harmful to human health. However, the significant computational requirements for dataset construction may exacerbate disparities between well-resourced and under-resourced research institutions, potentially widening existing gaps in research capabilities.h

# F DETAILED TRAINING AND INFERENCE ALGORITHM

## F.1 TRAINING PROCESS

---

**Algorithm 2** DEL-Ranking Training

---

**Require:** DEL dataset $\mathcal{D} = \{(f_i, p_i, M_i, R_i, y_i)\}_{i=1}^N$, where $f_i \in \mathbb{R}^d$ is molecular fingerprint, $p_i \in \mathbb{R}^m$ is binding pose, $M_i \in \mathbb{R}$ is matrix count, $R_i \in \mathbb{R}$ is target count, $y_i \in \{0, 1\}$ is functional group label

**Require:** Hyperparameters: $\rho$ (ranking weight), $\gamma$ (consistency weight), $\beta$ (PSR weight), $T$ (temperature), $\sigma$ (contrastive weight), $\tau$ (margin), learning rate $\eta$

**Ensure:** Trained model parameters $\theta$

 1: Initialize model parameters $\theta$ randomly
 2: **for** epoch = 1 to $N_{\text{epochs}}$ **do**
 3:     **for** each batch $\mathcal{B} \subset \mathcal{D}$ **do**
 4:         **// Forward Pass**
 5:         Extract fingerprint embeddings: $h_f = \text{FingerprintEncoder}(f_i)$
 6:         Extract pose embeddings: $h_p = \text{PoseEncoder}(p_i)$
 7:         Fuse representations: $x = \text{FusionModule}(\varsigma h_p + h_f)$
 8:         **// Chemical-Referenced Correction (CRC)**
 9:         $\hat{M}_i = \text{MatrixHead}(h_f)$          ▷ Predict matrix count from fingerprint only
10:         Initialize $\hat{R}_i \leftarrow 0, \hat{p}_i \leftarrow 0$
11:         **for** iteration $k = 1$ to $n_{\text{CRC}}$ **do**
12:             **if** use_feedback **then**
13:                 $\hat{R}_i \leftarrow [x; \hat{p}_i], \hat{p}_i \leftarrow [x; \hat{R}_i]$
14:             **else**
15:                 $\hat{R}_i \leftarrow x, \hat{p}_i \leftarrow x$
16:             **end if**
17:             $\hat{R}_i = \text{EnrichmentHead}(\text{ReadHead}(\hat{R}_i))$
18:             $\hat{p}_i = \text{ActHead}(\hat{p}_i)$
19:         **end for**
20:         **// Compute Losses**
21:         Compute ZIP loss: $\mathcal{L}_{\text{ZIP}} = -\sum_i \log[P(M_i|\lambda_M, \pi_M)] - \sum_j \log[P(R_j|\lambda_M + \lambda_R, \pi_R)]$
22:         **// Pairwise Soft Ranking Loss**
23: $$\mathcal{L}_{\text{PSR}} = -\sum_{i=1}^N \sum_{j \neq i, r_i > r_j} [\Delta_{ij} \cdot \sigma_{ij}(T)]$$
24:         where $\sigma_{ij} = \frac{1}{1 + e^{-|r_i - r_j|/T}}, \Delta_{ij} = \frac{\Delta \text{G}_{ij} \cdot \Delta \text{D}_{ij}}{Z}$
25:         **// Listwise Global Ranking Loss**
26: $$\mathcal{L}_{\text{LGR}} = -\sum_{i=1}^N \log \frac{\exp(\hat{r}_{\Omega(i)}/T)}{\sum_{j=i}^N \exp(\hat{r}_{\Omega(j)}/T)} + \sigma \sum_{i=1}^N \sum_{j>i} \mathcal{L}_{\text{con}}(\hat{r}_i, \hat{r}_j, \tau)$$
27:         **// Consistency Loss**
28:         $\mathcal{L}_{\text{consist}} = \|\hat{p}_i - y_i\| + \max\left(0, \|\hat{y}_i - \frac{\hat{r}_i}{\max_{i \in \{1,\dots,N\}} \hat{r}_i}\|_2^2 - \|y_i - \frac{r_i}{\max_{i \in \{1,\dots,N\}} r_i}\|_2^2\right)$
29:         **// Total Loss**
30:         $\mathcal{L}_{\text{total}} = \mathcal{L}_{\text{ZIP}} + \rho(\beta \mathcal{L}_{\text{PSR}} + (1 - \beta)\mathcal{L}_{\text{LGR}}) + \gamma \mathcal{L}_{\text{consist}}$
31:         **// Backward Pass**
32:         Compute gradients: $\nabla_\theta \mathcal{L}_{\text{total}}$
33:         Update parameters: $\theta \leftarrow \theta - \eta \nabla_\theta \mathcal{L}_{\text{total}}$
34:     **end for**
35: **end for**
36: **return** $\theta$

---

## F.2 INFERENCE PROCESS

---

**Algorithm 3** DEL-Ranking Inference

---

**Require:** Test compound with fingerprint $f_{\text{test}}$ and binding poses $p_{\text{test}}$
**Require:** Trained model with parameters $\theta$
**Ensure:** Predicted counts $\hat{M}_{\text{test}}$, $\hat{R}_{\text{test}}$ and activity probability $\hat{p}_{\text{test}}$

1:  **// Extract Features**
2:  $h_f = \text{FingerprintEncoder}_\theta(f_{\text{test}})$
3:  $h_p = \text{PoseEncoder}_\theta(p_{\text{test}})$
4:  $x = \text{FusionModule}_\theta(\varsigma h_p + h_f)$
5:  **// Predict Matrix Count**
6:  $\hat{M}_{\text{test}} = \text{MatrixHead}_\theta(h_f)$
7:  **// Predict Target Count and Activity**
8:  Initialize $\hat{R}_{\text{test}} \leftarrow 0$, $\hat{p}_{\text{test}} \leftarrow 0$
9:  **for** iteration $k = 1$ to $n_{\text{CRC}}$ **do**
10:     **if** use_feedback **then**
11:         $\hat{R}_{\text{test}} \leftarrow [x; \hat{p}_{\text{test}}]$, $\hat{p}_{\text{test}} \leftarrow [x; \hat{R}_{\text{test}}]$
12:     **else**
13:         $\hat{R}_{\text{test}} \leftarrow x$, $\hat{p}_{\text{test}} \leftarrow x$
14:     **end if**
15:     $\hat{R}_{\text{test}} = \text{EnrichmentHead}_\theta(\text{ReadHead}_\theta(\hat{R}_{\text{test}}))$
16:     $\hat{p}_{\text{test}} = \text{ActHead}_\theta(\hat{p}_{\text{test}})$
17:  **end for**
18:  **// Estimate Binding Affinity**
19:  Compute enrichment factor: $E_{\text{test}} = \frac{\hat{R}_{\text{test}}}{\hat{M}_{\text{test}}+\epsilon}$
20:  Rank compounds by $\hat{R}_{\text{test}}$ in descending order
21:  **return** $\hat{M}_{\text{test}}$, $\hat{R}_{\text{test}}$, $\hat{p}_{\text{test}}$

---

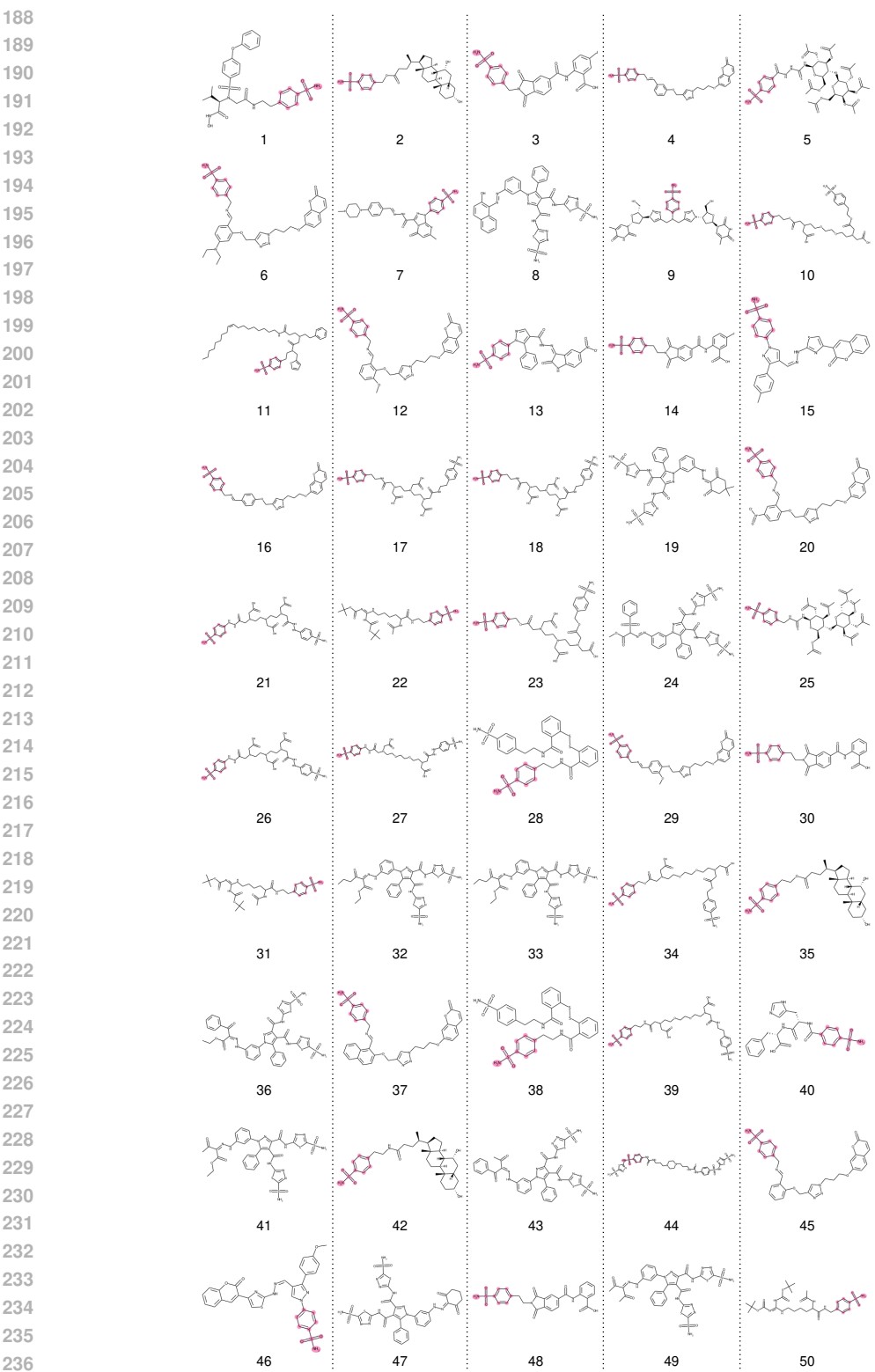

Figure 6: Visualization of the top-50 DEL-Ranking results on the 3p3h dataset. In molecules containing benzenesulfonamide, the benzenesulfonamide structure is highlighted.

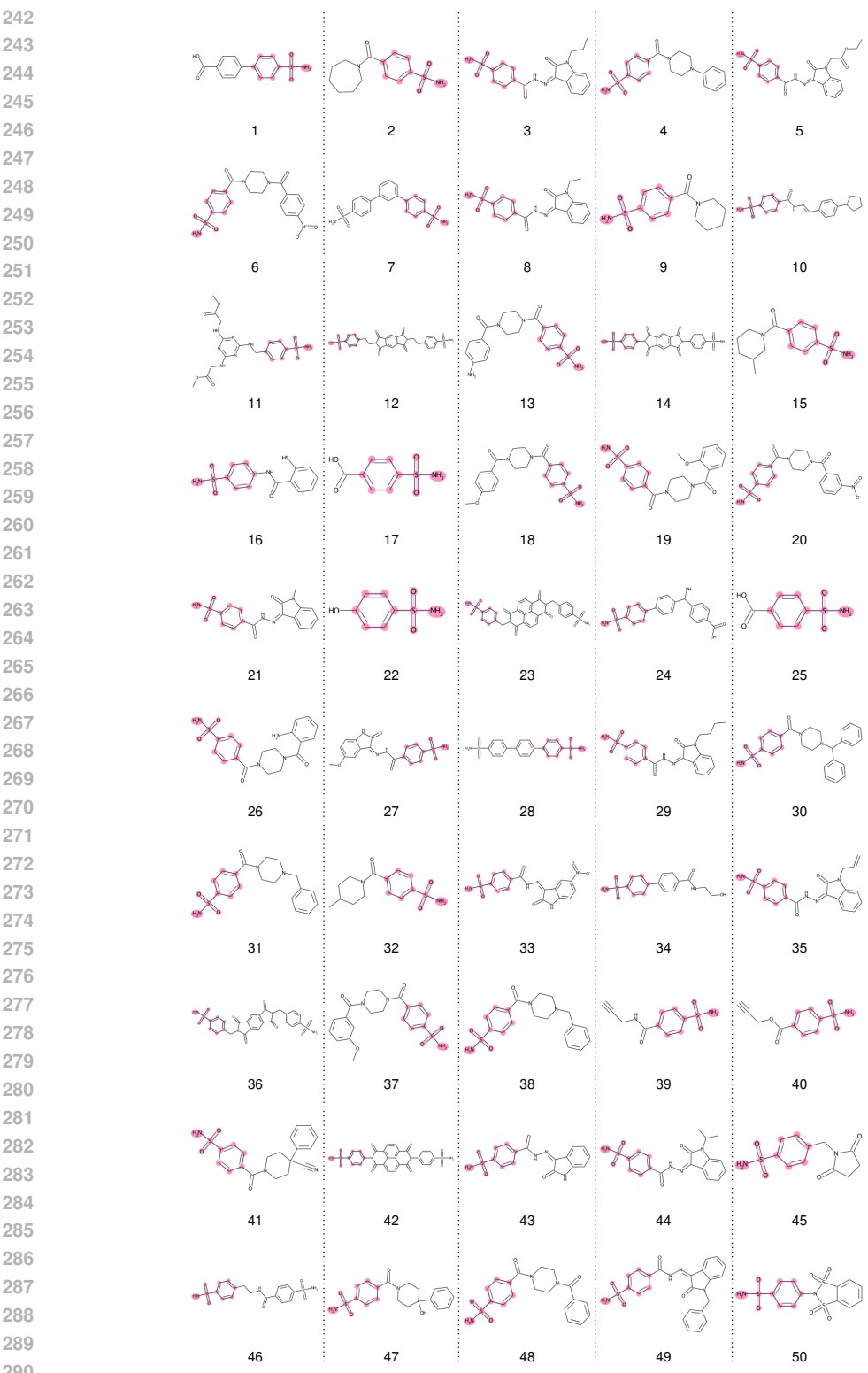

Figure 7: Visualization of the top-50 DEL-Ranking results on the 4kp5-A dataset. In molecules containing benzenesulfonamide, the benzenesulfonamide structure is highlighted.

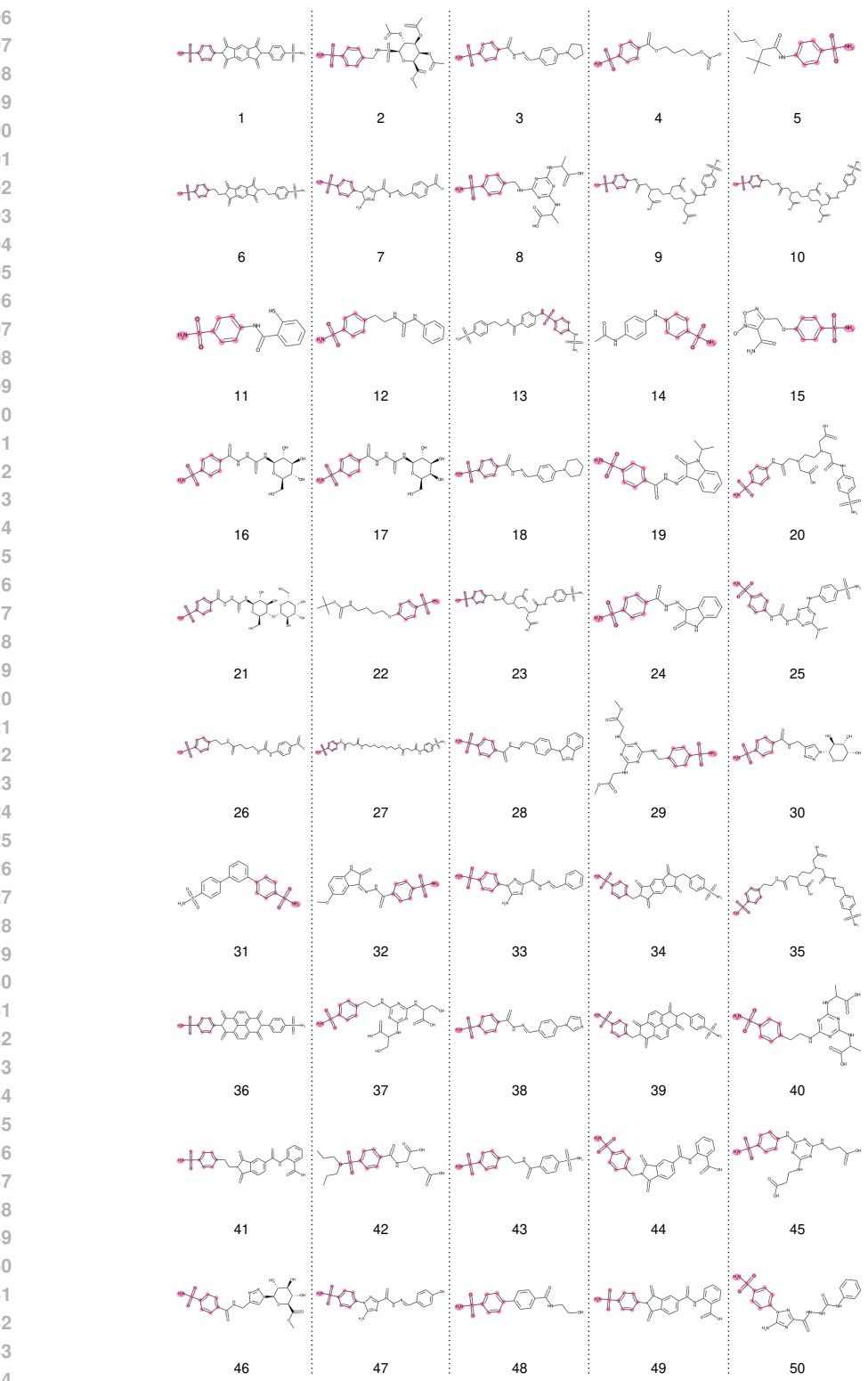

Figure 8: Visualization of the top-50 DEL-Ranking results on the 4kp5-OA dataset. In molecules containing benzenesulfonamide, the benzenesulfonamide structure is highlighted.

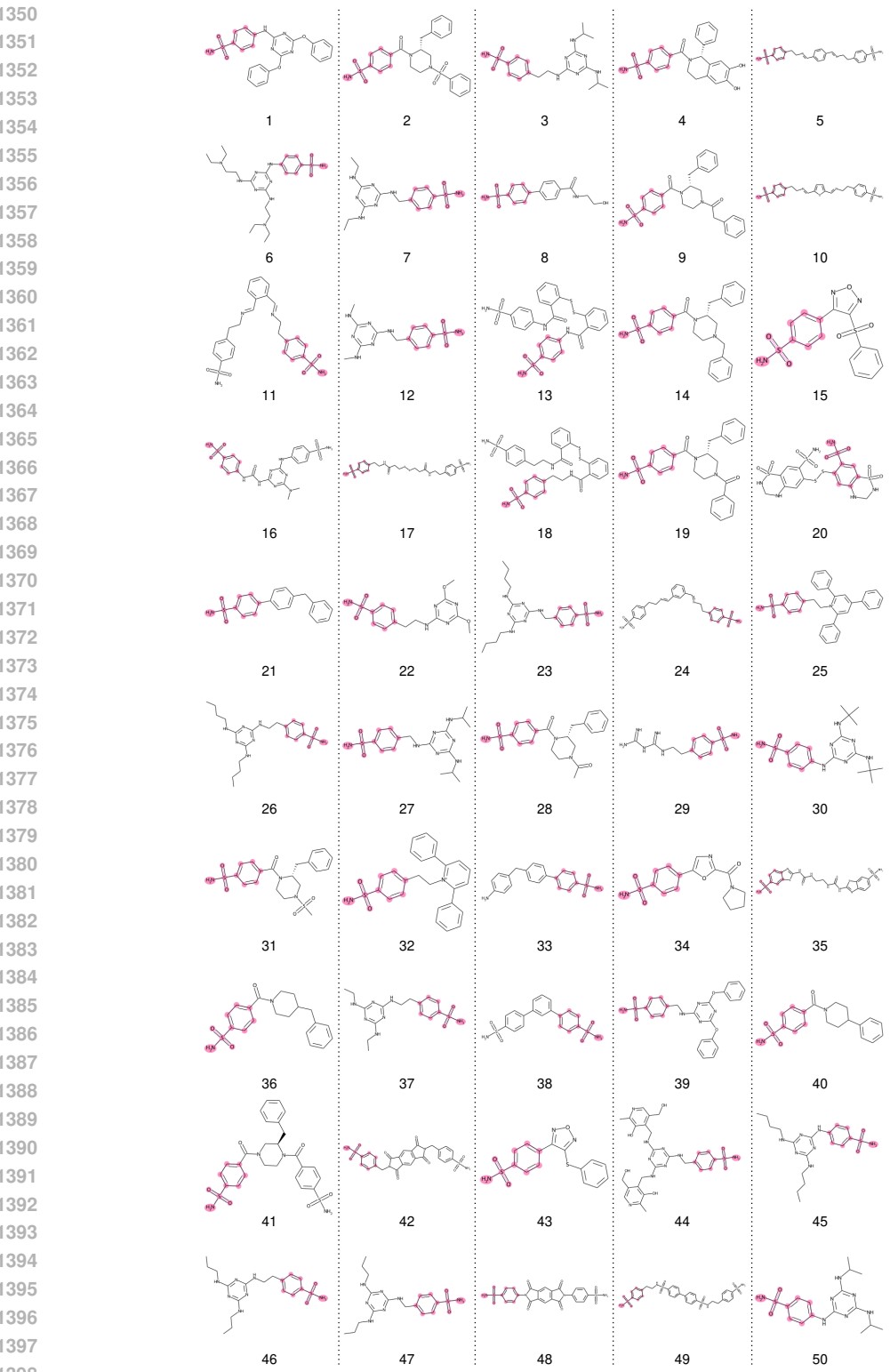

Figure 9: Visualization of the top-50 DEL-Ranking results on the 5fl4(9 pose) dataset. In molecules containing benzenesulfonamide, the benzenesulfonamide structure is highlighted.

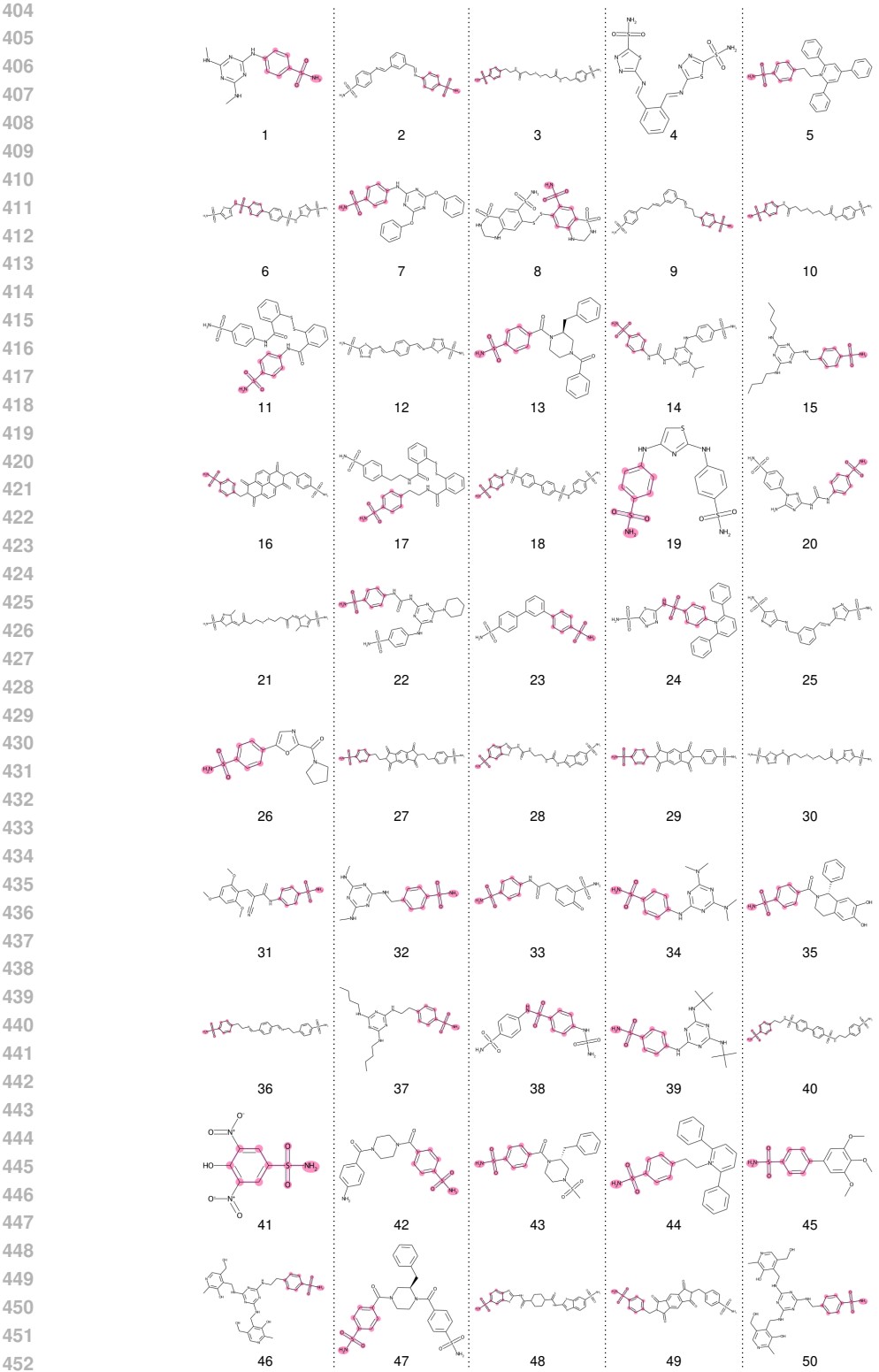

Figure 10: Visualization of the top-50 DEL-Ranking results on the 5fl4(20 pose) dataset. In molecules containing benzenesulfonamide, the benzenesulfonamide structure is highlighted.

# G    VISUALIZATION ON SELECTED CASES CONTAINING PYRIMIDINE SULFONAMIDE

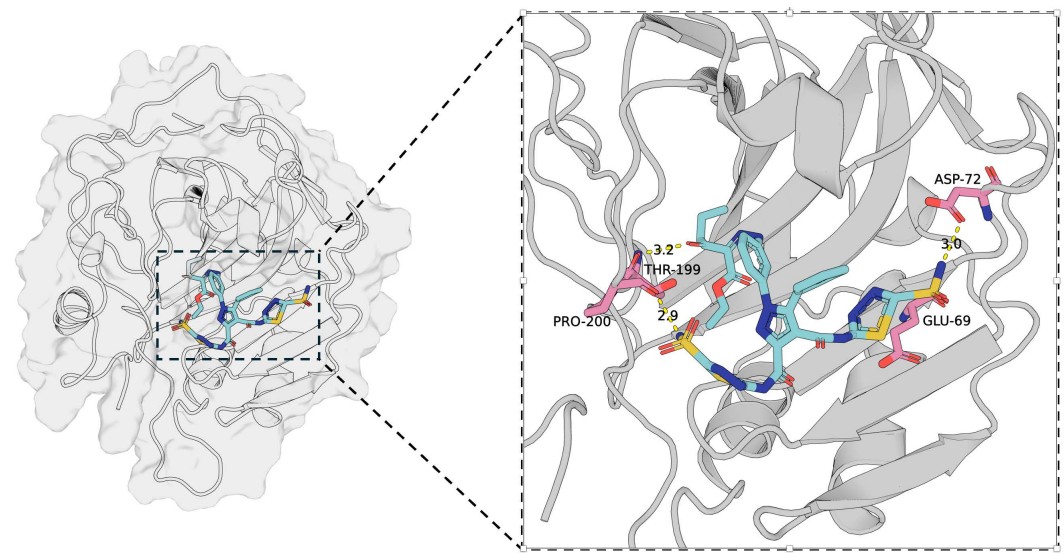

Figure 11: In 3p3h, THR199 likely forms hydrogen bonds with the ligand, while ASP72 and GLU69 participate in hydrogen bonding and electrostatic interactions. The corresponding $k_i$ value is 84.0.

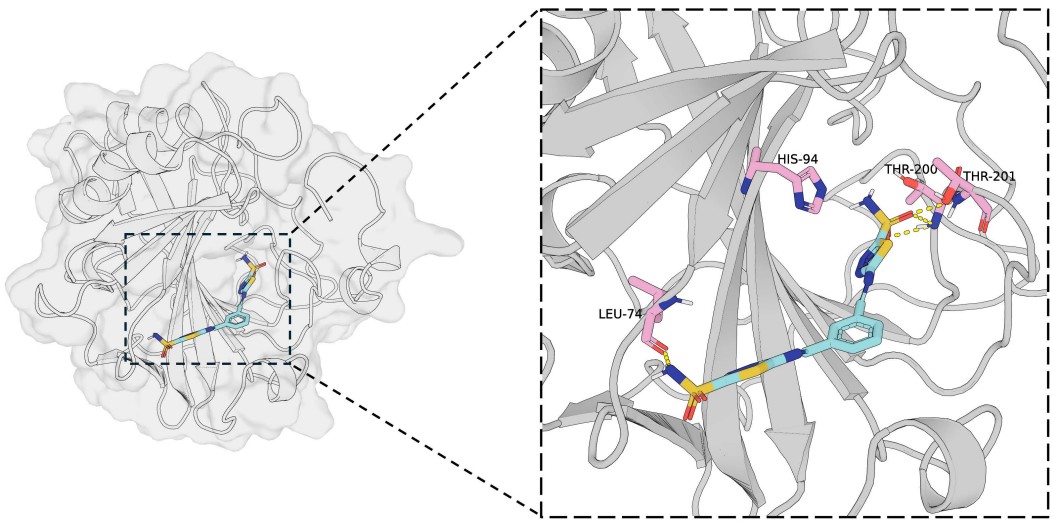

Figure 12: In 5fl4, LEU74 contributes through van der Waals forces or hydrophobic interactions, HIS94's imidazole side chain potentially forms hydrogen bonds, and THR201 engage in hydrogen bonding with the ligand. The corresponding $k_i$ value is 0.5.

# H    USAGE OF LANGUAGE MODELS

We use large language model (LLM) to aid in the preparation of this manuscript. Its use was limited to editorial tasks, including proofreading for typographical errors, correcting grammar, and improving the clarity and readability of the text.

