# OpenReview forum: "DEL-Ranking: Ranking-Correction Denoising Framework for Elucidating Molecular Affinities in DNA-Encoded Libraries"
_ICLR.cc/2026/Conference — ICLR 2026 Conference Desk Rejected Submission_

### Official Review · Reviewer_hriM · 2025-10-29

**Soundness:** 3
**Presentation:** 3
**Contribution:** 3
**Rating:** 6
**Confidence:** 2

**Summary:**

The paper targets two core issues in DEL screening—distribution noise in low-copy regimes and distribution shift between read counts and true affinities—by combining a ZIP-based count model with a dual-perspective ranking loss (pairwise + listwise) and a chemical-referenced correction (CRC) that aligns counts with functional-group activity signals. It further releases three datasets integrating 2D/3D structure, counts, and binary activity labels. Experiments across multiple CA targets show consistent gains in Spearman correlation (notably under high noise) and report interpretable insights, including robust detection of Benzene Sulfonamide and a novel Pyrimidine Sulfonamide motif.

**Strengths:**

The method is well-motivated by DEL-specific statistics (zero inflation, count noise) and integrates ranking objectives that stabilize relative orderings—a sensible choice for affinity prioritization. CRC injects chemical priors to reduce read count–affinity gaps while improving interpretability. The dataset contribution is valuable (2D/3D + activity labels) and strengthens reproducibility. Empirical evaluation is broad (multiple datasets, subset analysis, zero-shot generalization) and shows consistent improvements over strong baselines.

**Weaknesses:**

The framework is complex, with multiple losses, temperatures, margins, and weights, raising tuning cost and reproducibility concerns. CRC depends on functional-group labels that may be incomplete or noisy, risking bias or self-confirmation effects. The theoretical claims (information gain and expected error) are high level; practical conditions and assumptions (e.g., ZIP parameterization, batchwise Top-K normalization) are not fully stress-tested. The claim of correcting “low count but high activity” cases would benefit from more orthogonal experimental validation beyond DEL-derived signals.

**Questions:**

Could you quantify the coverage and noise rate of functional-group labels used by CRC, and report robustness when labels are missing or corrupted? How sensitive is performance to key hyperparameters (temperature T, margin τ, Top-K normalization K, and loss weights ρ/γ), and can you provide default heuristics and ablation curves? How are ZIP parameters (π, λ) tied to experimental factors (library copy distributions, amplification cycles); did you compare shared vs separate or hierarchical parameterizations? For the Pyrimidine Sulfonamide motif, do you have external validation (independent assays or public Ki) and structural evidence (docking/MD/contact analysis) distinguishing its binding mode from Benzene Sulfonamide?

---

> ### Author Response · Authors · 2025-11-27
>
> Dear Reviewer,
>
> Thank you very much for your insightful suggestions and reviews of our manuscript. We found that most of them are constructive and helpful.
>
> **WEAKNESS1 and QUESTION2 (Complexity, multiple losses, hyperparameters, reproducibility).**
> > Conceptually, DEL-Ranking uses a simple three-term objective—ZIP loss + ranking loss + optional CRC consistency loss (Eq. 11)—and in practice only a small subset of hyperparameters is tuned ($\rho, \gamma$, ranking temperature $T$, margin $\tau$, Top-K $K$, and the structure/fingerprint balance). All other architectural choices are fixed across datasets. We already provide systematic ablations over these key hyperparameters in Appendices C.1–C.2 and Table 4, showing that performance varies smoothly and remains strong over a broad range of settings rather than relying on a single narrow optimum.
>
> **WEAKNESS2 and QUESTION1 (CRC dependence on functional-group labels).**
> > The ablation results were shown in Table 3. CRC is an optional module that is only activated when coarse binary activity labels are available; on label-free datasets (5fl4-9p/20p) $\gamma$ is set to 0, so the reported gains there come solely from ranking + ZIP. When labels exist, they are defined by simple substructure rules (presence of the arylsulfonamide building block BB3-197 for CAS-DEL CA2/CA12 [1], a known privileged motif for carbonic anhydrase inhibitors) rather than model-dependent scores, which limits self-confirmation.
>
> **WEAKNESS3 (High-level theory).**
> > Our theoretical results (Lemma 3.1, Theorem 3.2) are intentionally stated at the level of expected risk for a generic ZIP + ranking combination, in line with prior work on LambdaRank/NDCG-style objectives where the scalar loss is defined implicitly through its gradient [2]. Making all assumptions fully DEL-specific (e.g., explicit copy-number distributions and amplification cycles) would require strong system-dependent modeling choices and is therefore beyond the scope of this first study;
>
> **QUESTION3: Relationship between ZIP parameters ($\pi, \lambda$) and experimental factors;**
> > In our implementation, $\pi$ and $\lambda$ are per-compound outputs of the neural network for both target and matrix counts, i.e., they are learned functions of molecular features and pose-level structural information, not fixed hyperparameters. This allows the model to implicitly absorb experimental factors such as initial copy numbers and amplification efficiency without requiring explicit parameterization for each selection protocol, which is consistent with other DEL ZIP-based models [3].
>
> **QUESTION4: External validation and structural evidence for the Pyrimidine Sulfonamide motif.**
> > The testing data (including $K_i/K_d$) from ChembL is the one validated in the wet-lab experiments. For pyrimidine sulfonamide, we performed two types of orthogonal validation beyond DEL counts: (i) independent biochemical $K_i$ measurements for ten DEL-Ranking-prioritized hits (five from 3p3h, five from 5fl4) containing pyrimidine sulfonamide, which show $K_i$ values comparable to or better than canonical benzene-sulfonamide analogs (Figure 3 and Sec. 4.2), and (ii) docking/interaction analysis of representative pyrimidine- vs benzene-sulfonamide complexes, where both motifs form similar sulfonamide–Zn coordination and hydrogen-bond networks in the CA active site.
>
> **Reference:**
>
> [1] Hou, R., Xie, C., Gui, Y., Li, G., & Li, X. (2023). Machine-learning-based data analysis method for cell-based selection of DNA-encoded libraries. ACS omega, 8(21), 19057-19071.
>
> [2] Ballester, P. J. (2019). Selecting machine-learning scoring functions for structure-based virtual screening. Drug discovery today: technologies, 32, 81-87.
>
> [3] Lim, K. S., Reidenbach, A. G., Hua, B. K., Mason, J. W., Gerry, C. J., Clemons, P. A., & Coley, C. W. (2022). Machine learning on DNA-encoded library count data using an uncertainty-aware probabilistic loss function. Journal of chemical information and modeling, 62(10), 2316-2331.
>
> Best regards,
>
>
> Authors

---

### Official Review · Reviewer_aejq · 2025-11-01

**Soundness:** 2
**Presentation:** 2
**Contribution:** 2
**Rating:** 2
**Confidence:** 4

**Summary:**

DEL-Ranking is a novel model for training on DEL data that incorporates new loss functions to improve molecule ranking. The model is built on DEL-Dock, receiving input that includes the chemical structure and binding pose of the ligand. The key contribution is the development of both local and global ranking losses tailored for ranking molecules in noisy DEL screen data. The pairwise soft ranking loss models relationships between compounds with noisy count data, while the listwise global ranking loss helps to identify excessive zero read counts. To address distribution shifts, another loss is created based on functional group labels. DEL-Ranking outperforms other baselines across two carbonic anhydrase datasets, measured by Sp and SubSp metrics (Spearman correlation on dataset subsets). The model has the potential to enhance the efficiency of DEL-based hit discovery screening.

**Strengths:**

- The proposed ranking losses are well-designed and based on effective strategies from other domains (like document ranking).
- The experimental results are strong, as DEL-Ranking outperforms other baselines across two carbonic anhydrase targets.
- The results of the model can be used to discover new functional groups that correlate with experimental activity, e.g. pyrimidine sulfonamide as a carbonic anhydrase inhibitor.
- An ablation study is conducted to examine the importance of each loss function term.
- Given the strong performance in the benchmarks, DEL-Ranking is well poised to improve the effectiveness of DEL-based hit discovery campaigns.

**Weaknesses:**

- The functional group labels or binary affinity labels derived from functional group analysis are not defined in the paper. How are these labels computed? Are they specific to the target, such as benzene sulfonamides being important for carbonic anhydrases?
- I do not understand the importance of Theorem 3.2. The text says “The combined loss function consistently outperforms the standard ZIP approach in expected performance.” However, the ZIP loss and the rank loss are two completely different loss functions, so comparing their values is not exactly fair. Following that logic, we could imply that the best loss function would be a constant zero function.
- In Equation 5, the inequality $r_i > r_j$ may indicate a problem with the loss function. Is it necessary if the expression being summed is perfectly symmetrical? $\sigma_{ij}$ is obviously symmetrical and $Z$ is constant (always based on top predictions). The $\Delta$ terms are not symmetrical, but they are multiplied by each other, making the product symmetrical. Why not use $j>i$ instead of $r_i > r_j$ then?
- If in Equation 9, the sum for the con loss is over $j \neq i$, and the margin of $\tau$ is enforced for $i > j$, then the loss for $j > i$ would be huge (in fact, $f(\hat{r}_i) - f(\hat{r}_j)$ should be negative).
- The loss functions in Algorithm 2 are different than the ones defined in the paper.
- The code is not shared, and there is no reproducibility section that would explain whether the code will be published upon the paper acceptance. The lack of available implementation makes it impossible to verify the actual formula for the proposed loss functions (the core contribution), given the inconsistencies indicated above.

**Questions:**

1. How can the model be trained with the PSR loss if the predictions are only included in the normalization term $Z$? Moreover, the loss function applies only to the top K predictions. Or maybe the predicted ranks in Equation 7 are also differentiable (if so, how)? The other parts of the loss function use the true read counts, rather than the predicted ones.
2. If $\Omega_i$ is the rank of all compounds, not only the top ones, would that not cause issues with the compounds with near-zero counts, which are ordered randomly due to the experimental noise?
3. Could you explain how Equation 14 corresponds to the PSR loss? Moreover, the terms in the first parenthesis should cancel out given the symmetry of $\Delta$.

---

> ### Author Response · Authors · 2025-11-27
> **Response to reviewer aejq (1/2)**
>
> Dear Reviewer,
>
> Thank you very much for your insightful suggestions and reviews of our manuscript. We found that most of them are constructive and helpful.
>
> **WEAKNESS1 (functional-group / activity labels are not defined):**
> > We have stated it in the Section 4 Dataset paragraph in the initial draft. Our functional labels are target-specific and derived directly from known CA pharmacophores in the underlying DEL designs. For the CAS-DEL CA2/CA12 datasets, the binary label $y_i$ is set to 1 if and only if the third-cycle building block BB3-197 (an arylsulfonamide/benzene sulfonamide) is present in the molecule, following the CAS-DEL design where this motif is the canonical CA binder [1]. For the CA9 (5fl4) datasets, such labels are unavailable and CRC is not used (only the ranking + ZIP losses are active).
>
> **WEAKNESS2 (importance and interpretation of Theorem 3.2; “constant zero loss” concern):**
> > We apologize for the imprecise wording around Theorem 3.2. The theorem does not compare raw pointwise values of $L_{ZIP}$ and $L_{rank}$, but the expected regression risk of the combined model $L_C = \alpha L_{ZIP} + (1-\alpha)L_{rank}$ relative to a model trained with $L_{ZIP}$ alone.
> > Because any non-trivial ranking loss satisfies $\mathbb{E}[L_{rank}(r_{ZIP}, R)] > 0$, a constant zero predictor cannot minimize $L_C$; it incurs large ranking loss even if its ZIP term is small. The proof in Appendix A.1 explicitly uses the information-gain lemma (Lemma 3.1) together with this positivity to bound $\mathbb{E}[L_C(r_C)] < \mathbb{E}[L_{ZIP}(r_{ZIP})]$; we have revised the text around Theorem 3.2 to emphasize that the claim is about expected combined loss, not about direct comparability of heterogeneous losses, and explicitly mention that constant predictors are excluded because they violate the ranking objective.
>
> **WEAKNESS3 (Equation 5 and the condition $r_i > r_j$):**
> > You are correct that the term $\lambda_{ij}$ is symmetric in $(i,j)$ once $\lambda_{ij} = \lambda_{ji}$ is defined via the NDCG gain/discount factors.
> > The inequality $r_i > r_j$ in Eq. (5) is not required by the theory; it is a practical way to avoid double-counting each unordered pair $\{i,j\}$. In the implementation we iterate over unordered pairs, which is equivalent to summing over $i < j$. We need to claim explicitly that the sum is over unordered pairs, and clarify that this does not affect gradients or the learned ranking.
>
> **WEAKNESS4 (Equation 9 and the contrastive term; concern about large loss for $j > i$):**
> > We greatly appreciate you pointing out this detail. To clarify the intent of the formula, we have revised the summation condition in the contrastive loss from $j \neq i$ to $j > i$, as shown in Equation 18 of the updated draft. This revision eliminates potential confusion caused by the previous notation. In our implementation, the contrastive loss $\mathcal{L}_{\mathrm{con}}(\hat{r}_i, \hat{r}_j, \tau)$ is evaluated once per unordered pair ${i, j}$; we do not simultaneously enforce a positive margin for both $i > j$ and $j > i$. The margin condition is applied based on the sorted scores $f(r_i)$ and $f(r_j)$, and the loss is zero whenever their difference already exceeds $\tau$.
>
> **WEAKNESS5 (inconsistency between Eq. (5), Algorithm 2, and Eq. (14)):**
> > Thank you for pointing out this formatting detail. The issue was caused by the display settings in the Algorithm section. We have updated it and ensured that the loss functions are now consistent with those in the main text.
>
> **WEAKNESS6 (code availability and reproducibility):**
> > We agree that reproducibility is crucial, especially given the complexity of the loss functions. We have prepared an anonymized repository containing
> > 1. full training and evaluation code implementing the corrected PSR/LGR objectives,
> > 2. configuration files and random seeds that reproduce every number in Tables 1–4,
> > 3. split indices for each dataset.
> > 4. the raw predictions and experimental results are of huge amount. We feel so sorry that we cannot merge and sort them in the limited time. But we are happy to provide any if you would like to see some of them. Many thanks for your understanding!
> >
> > The GitHub repo can be found at: https://anonymous.4open.science/r/del_ranking_2333.

---

> ### Author Response · Authors · 2025-11-27
> **Response to reviewer aejq (2/2)**
>
> **QUESTION1: How can the model be trained with PSR loss if predictions appear only in the normalization term $Z$? Are the ranks differentiable?**
> > We have updated the modeling of LambdaRank and ListMLE in the revised draft. After correcting Eq. (5) as described above, predictions $r_i$ influence PSR through both $\lambda_{ij}(r_i - r_j)$ and the top-K normalization $Z(r)$, not just $Z$.
> > The gradient in Eq. (14) makes this explicit: $\partial L_{PSR}/\partial r_i$ depends on $\lambda_{ij}$, $\lambda_{ji}$, and their derivatives, all functions of $r_i - r_j$.
> > As in LambdaRank, the permutation is treated as a fixed ordering within each forward pass; we do not backpropagate through the discrete ranks themselves, only through the continuous scores. This is standard practice in learning-to-rank methods that optimize NDCG surrogates [2].
>
> **QUESTION2: Why not restrict $i$ to the top compounds only, given noisy low-count items?**
> > Following the response to the last question, we use the full permutation but the contribution of very low-rank compounds is strongly down-weighted. In PSR, the NDCG-style term $D_{ij}$ suppresses contributions from swaps involving only low-rank items, and in LGR the listwise log-likelihood is dominated by higher-ranked terms in the denominator.
> > In addition, the normalization $Z$ only aggregates the top-K scores, so the overall scale of the PSR gradient is governed by high-score items.
>
> **QUESTION3: How does Equation 14 correspond to the PSR loss? Why don’t the terms cancel when $\lambda_{ij} = \lambda_{ji}$?**
> > Eq. (14) is derived from a logistic pairwise objective where each pair $(i,j)$ is weighted by its NDCG impact $\lambda_{ij}$, exactly as in LambdaRank’s “lambda” gradients [3]. While $\lambda_{ij} = \lambda_{ji}$ by construction, the sigmoid terms are not symmetric: $\sigma_{ij}(r_i - r_j) \neq \sigma_{ji}(r_i - r_j)$ and their derivatives have opposite signs. Consequently, the two brackets in Eq. (14) combine into a non-zero quantity proportional to $\lambda_{ij}[\sigma_{ij} - \sigma_{ji}]$ plus additional terms involving $\partial \lambda_{ij}/\partial r_i$ and $\partial \lambda_{ji}/\partial r_i$.
> > This is precisely the mechanism that encourages the model to increase the score of items whose upward swaps would most improve NDCG.
>
> **Reference:**
>
> [1] Hou, R., Xie, C., Gui, Y., Li, G., & Li, X. (2023). Machine-learning-based data analysis method for cell-based selection of DNA-encoded libraries. ACS omega, 8(21), 19057-19071.
>
> [2] Burges, C., Shaked, T., Renshaw, E., Lazier, A., Deeds, M., Hamilton, N., & Hullender, G. (2005, August). Learning to rank using gradient descent. In Proceedings of the 22nd international conference on Machine learning (pp. 89-96).
>
> [3] Burges, C., Ragno, R., & Le, Q. (2006). Learning to rank with nonsmooth cost functions. Advances in neural information processing systems, 19.
>
> Best regards,
>
> The Authors

---

### Official Review · Reviewer_YDEk · 2025-11-03

**Soundness:** 3
**Presentation:** 3
**Contribution:** 3
**Rating:** 8
**Confidence:** 3

**Summary:**

This paper addresses two fundamental challenges in DNA-encoded library (DEL) screening: (1) stochastic noise from Poisson fluctuations in low copy number regimes, and (2) systematic biases between read counts and binding affinities. The authors propose DEL-Ranking, which combines a dual-perspective ranking mechanism (Pairwise Soft Rank + Listwise Global Rank) with a Chemical-Referenced Correction (CRC) module that leverages functional group information. The paper also contributes three novel DEL datasets with multi-modal data (2D sequences, 3D conformations, functional labels) and reports up to 28% improvement in Spearman correlation under high-noise conditions.

**Strengths:**

The paper addresses a well-motivated and important problem in drug discovery with strong domain grounding. The theoretical foundation is solid, with Lemma 3.1 and Theorem 3.2 providing formal justification for why ranking losses provide information gain over ZIP alone through an information-theoretic perspective. The methodology is comprehensive, with the dual approach addressing both distribution noise via ranking and distribution shift via CRC being well-designed. The combination of local (PSR) and global (LGR) ranking perspectives is principled and theoretically justified.

A significant contribution is the release of three novel DEL datasets with 2D sequences, 3D conformations, and functional labels, which addresses a critical resource gap in the community. The experimental validation is thorough, spanning 5 datasets with proper error bars, ablations, zero-shot evaluation, and extensive hyperparameter analysis. The appendix provides comprehensive visualizations and detailed algorithms. The identification of Pyrimidine Sulfonamide as a novel binding motif beyond known Benzene Sulfonamide groups demonstrates practical interpretability. Results show consistent state-of-the-art performance across multiple diverse datasets, with particularly strong improvements in high-noise scenarios like 4kp5-OA.

**Weaknesses:**

The CRC module requires prior knowledge of affinity-determining functional groups, which the authors acknowledge is "frequently unavailable in many DEL datasets." The 5fl4 datasets lack functional group labels, yet no performance is reported for DEL-Ranking without CRC on these datasets. Table 3's ablation only shows "w/o CRC" on datasets that have labels. This creates a circular dependency: you need to know binding-relevant groups to discover binding-relevant groups. The paper must explicitly demonstrate how DEL-Ranking performs on label-free datasets (5fl4) without CRC to establish whether the ranking mechanism alone provides value over DEL-Dock.

Table 4 reveals ranking weight ρ varies by 2-3 orders of magnitude (1e8 to 1e11) and temperature T ranges from 0.2 to 0.9 across datasets. The authors acknowledge this requires "dataset-specific hyperparameter optimization," risking overfitting and making new dataset application impractical. Table 2's zero-shot results show DEL-QSVR sometimes outperforms DEL-Ranking (4kp5-OA), suggesting simpler methods may generalize better despite added complexity. The method needs principled hyperparameter selection guidelines or automatic tuning procedures.

**Questions:**

This paper makes solid contributions to an important problem in drug discovery. The ranking-based approach is well-motivated with sound theoretical foundation, the dataset contribution is valuable, and the experimental validation is generally thorough. However, three critical limitations significantly reduce practical applicability: the severe dependency on functional group labels (with no demonstration of value when unavailable), unproven scalability to real-world library sizes, and hyperparameter sensitivity that threatens generalization. The magnitude of improvements is sometimes modest and may not justify the added complexity.

---

> ### Author Response · Authors · 2025-11-27
>
> Dear Reviewer,
>
> Thank you very much for your insightful suggestions and reviews of our manuscript. We found that most of them are constructive and helpful. The followings are our response:
>
> **WEAKNESS1 and QUESTION1 (Dependence on functional-group labels and CRC).**
> > CRC is an optional module that is only activated when functional-group labels are available; on label-free datasets such as 5fl4 we already train with γ=0, i.e., DEL-Ranking reduces to the ranking+ZIP model without CRC. We did make this explicit in Sec. 3.3/4.1, the ranking-only model (no CRC) still outperforms DEL-Dock, clarifying that the ranking mechanism itself provides value even without labels.
>
> **WEAKNESS2 (Availability of functional groups in practice).**
> > We agree that fully annotated affinity-determining groups are not always available. In practice, CRC only needs coarse binary labels for one or a few privileged motifs per target, which can be obtained automatically from SMILES via substructure queries or from prior SAR knowledge.
>
> **WEAKNESS3 and QUESTION2 (Hyperparameter sensitivity).**
> > The large numerical range in Table 4 mainly reflects different scales of LZIP vs Lrank (see ρ) across datasets; in all cases we choose ρ and T by a simple heuristic that matches gradient magnitudes of LZIP and Lrank on a small validation batch. We have already described this recipe in Appendix C.2.
>
> **WEAKNESS4 (Scalability and practical benefit vs simpler methods).**
> > Training and inference scale linearly with the number of compounds because ranking and CRC are computed within mini-batches with top-K truncation; empirically our datasets (≈8×10⁴–10⁵ unique molecules) are already comparable to typical deduplicated DEL campaigns, and training completes comfortably on a single GPU. We have added a runtime/scaling table in Appendix D.4. Also, we emphasized that the largest gains of DEL-Ranking occur in the high-noise 4kp5-OA setting and in functional-motif discovery (pyrimidine sulfonamide), where simpler models like DEL-QSVR cannot provide comparable interpretability in the initial draft.
>
> **QUESTION3 (Apply to real-world library sizes in practice).**
> > Real-world DEL campaigns typically downsample to tens of thousands of unique barcodes for modeling after aggregation and filtering, which matches the scale of our experiments; the ranking loss operates on batches, so memory usage is independent of total library size.
>
> **QUESTION4 (Modest improvement)**
> > We acknowledge that the read-count distribution of the dataset itself can cause performance fluctuations, and some improvements brought by DEL-Ranking may not pass the significance test. However, this does not affect the effectiveness of ranking in DEL denoising, especially on challenging datasets like 4kp5-OA that current methods cannot handle.
>
> Best regards,
>
> The Authors

---

### Official Review · Reviewer_Gy5j · 2025-11-11

**Soundness:** 3
**Presentation:** 2
**Contribution:** 3
**Rating:** 4
**Confidence:** 4

**Summary:**

This paper introduces DEL-Ranking, a framework designed to improve affinity inference from DNA-Encoded Library (DEL) read counts by addressing two persistent issues: stochastic noise from low-copy regimes and systematic distribution shifts between observed read counts and true binding affinities. The method combines (i) a dual-perspective ranking loss—Pairwise Soft Rank (PSR) and Listwise Global Rank (LGR)—to stabilize ordering of read counts, (ii) a Chemical-Referenced Correction (CRC) module that leverages binary functional-group labels to refine predictions, and (iii) ZIP modeling for target and matrix counts. Across five CA2/CA9/CA12 datasets, the approach reportedly achieves up to ~28% improvement in Spearman correlation under high-noise conditions and identifies a new functional motif (pyrimidine sulfonamide) beyond the known benzenesulfonamide scaffold.

**Strengths:**

**Origanality:** The paper proposes a novel integration of ranking-based learning (PSR + LGR) with ZIP modeling and CRC. This unified formulation appears new in the DEL domain and is conceptually strong.

**Quality:** The mathematical formulation is coherent and supported by ablation studies that clarify the contribution of each component. The dual-ranking objective is well-motivated and theoretically sound.

**Clarity:** Most equations and concepts are clearly presented. The overall flow of the methodology is understandable and well-grounded in prior DEL and machine learning literature.

**Significance:** If reproducible, the demonstrated improvements in rank-correlation and functional motif discovery could make this a valuable tool for prioritizing compounds in virtual screening workflows.

**Weaknesses:**

**Origanality:** While the ranking approach is well-applied, PSR and LGR are derived from classic Learning-to-Rank methods (e.g., LambdaRank, ListMLE). The paper would benefit from a clearer discussion of how these are adapted or extended for the DEL setting.

**Quality:**

- **Runtime and efficiency:** The paper does not provide wall-clock or throughput benchmarks against baselines (e.g., DEL-Dock, MLP-ZIP). Given the computational demands mentioned in the Limitations section, this omission makes it hard to assess practical applicability.

- **Data leakage risk:** The dataset construction procedure is described, but no details are given about time-based or scaffold-based data splitting, nor any similarity filtering (e.g., Murcko or Tanimoto). Without this, data leakage cannot be ruled out.

- **Reproducibility:** Code, datasets, and raw results are not provided.

**Clarity:** Figures could be improved with clearer legends and consistent labeling. Indices (i, j) are undefined in the ZIP objective. A simple illustrative example contrasting PSR and LGR would help readers grasp their differences.

**Significance:** Without public code or raw results (predictions, splits, configurations), it is difficult for the community to verify or build upon this work.

**Adjustments and Suggestions**

- Include runtime and efficiency tables for both training and inference.
- Describe data split strategies clearly, with scaffold/time-split justification and leakage analysis.
- Improve figure legends and notation consistency.
- Release code, raw predictions, and dataset splits with configurations and seeds to enable full reproducibility.

If the authors can provide (a) training/inference code, (b) raw prediction outputs, (c) split files with time/scaffold policies, (d) configuration and seed details, and (e) reproducibility instructions, I would be happy to reconsider and potentially raise my rating.

**Questions:**

**1.** Could you report the training and inference runtimes (e.g., per 100k compounds) compared to DEL-Dock or MLP-ZIP on the same hardware? The manuscript refers to acceleration, but no quantitative evidence is presented.

**2.** How were the train/validation/test splits constructed? Were time-based or scaffold-based splits applied?

**3.** Baseline accuracy appears to collapse on the 4kp5-OA dataset (Figures 3–4). Can you provide an analysis of this distribution shift, such as read count inflation or pose variance, and explain why DEL-Ranking maintains stability there?

**4.** How does the CRC refinement differ from standard iterative-refinement methods? Any ablation or failure-mode analysis would be useful.

---

> ### Author Response · Authors · 2025-11-27
>
> Dear Reviewer,
>
> Thank you very much for your insightful suggestions and reviews of our manuscript. We found that most of them are constructive and helpful.
>
> **WEAKNESS1 (Originality of PSR/LGR vs classic Learning-to-Rank).**
> > We have added preliminaries of ranking functions to the revised manuscript (section 3.1), together with our improvements (Section 3.2). Our PSR and LGR losses are adapted to DEL read-count data rather than direct copies of LambdaRank/ListMLE: PSR uses continuous read counts with NDCG-style gain/discount terms and a top-K normalization to handle zero-inflated batches, while LGR augments ListMLE with a contrastive margin loss specifically designed to separate zero/near-zero read counts.
>
> **WEAKNESS2 and QUESTION1 (Runtime and efficiency).**
> > We agree that wall-clock measurements are important. We have run additional benchmarks and have added a table in Appendix D.4 reporting training time per epoch and inference throughput (time per 100k compounds) for DEL-Ranking, DEL-Dock, and MLP-ZIP on the same hardware (single RTX 3090 + 32-core CPU). These results show that DEL-Ranking has comparable training cost to DEL-Dock and similar inference throughput to MLP-ZIP, while providing higher Spearman correlation. We need to clarify that the most expensive step is one-time offline dataset construction (docking), which is shared across pose-based methods.
>
> **WEAKNESS3 and QUESTION2 (Data leakage risk / split strategy).**
> > We apologize for the lack of detail on splits. DEL-Ranking is trained exclusively on DEL read counts and evaluated on an external ChEMBL dataset that shares no molecules with the training libraries, so the reported Spearman correlations are computed on ligands that are fully disjoint from the DEL training data. Within each DEL dataset, we use molecule-level random splits (train/validation/test with a fixed ratio and fixed random seed) and apply the same splits to all baselines. In the revised manuscript we have described this protocol in Sec. 4 (Data paragraph) and released the exact split index file in the anonymous code repo: https://anonymous.4open.science/r/del_ranking_2333.
>
> **WEAKNESS4 (Reproducibility: code, configs, raw outputs).**
> > We fully agree about the importance of reproducibility. We have prepared an anonymized repository containing https://anonymous.4open.science/r/del_ranking_2333:
> > 1. full training and inference code for DEL-Ranking,
> > 2. configuration and seed files for all experiments,
> > 3. the readme instructions for re-production,
> > 4. split indices for each dataset,
> > 5. the raw predictions and experimental results are of huge amount. We feel so sorry that we cannot merge and sort them in the limited time. But we are happy to provide any if you would like to see some of them. Many thanks for your understanding!
>
> **WEAKNESS5 (Clarity: figures, notation, and PSR vs LGR explanation).**
> > Thanks for pointing out the details. In the revised draft, we have updated the ranking function preliminaries in Section 3.1 to make it more readable and easy to understand. Also, in Section 3.2, we had mentioned the differences between our proposed methods and the existing ranking functions to boost the understanding.
>
> **QUESTION3 (Performance differences across datasets):**
> > This was reflected in our hyperparameter and structure-scaling ablations in Appendix D.1–D.2 already. The 4kp5-OA dataset corresponds to an overexpression condition and exhibits a much more skewed and noisy read-count distribution: compared to 3p3h and 4kp5-A, it has a heavier tail, a larger fraction of saturated high counts, and stronger zero-inflation. Baseline ZIP or enrichment-based models operate mainly on absolute counts and are therefore sensitive to this distribution shift, leading to unstable regression and degraded Spearman correlation. DEL-Ranking mitigates this by optimizing relative orderings via PSR/LGR, which down-weights unstable absolute magnitudes, and using CRC to align read counts with functional-group activity labels.
>
> **QUESTION4 (CRC refinement vs iterative-refinement)**
> > CRC is distinct from generic self-training or iterative-refinement schemes in two ways: (i) it couples two prediction heads—read counts and functional-group activity—through a consistency loss that explicitly aligns normalized read counts with activity probabilities, and (ii) it operates on fused 2D/3D embeddings with a fixed small number of refinement iterations, using functional-group labels as an external chemical “reference” to correct read-count biases.
> > We have clarified this in Sec. 3.3 and contrast CRC with standard self-training. In addition to the existing ablation on structure scaling (Appendix D.2). The ablation results can be referred to in Table 3.
>
> Best regards,
>
> Authors

---

### Author Response · Authors · 2025-11-27
**Global response to all reviewers**

Dear Reviewers,

We genuinely appreciate the time and effort you dedicated to reviewing our manuscript. Your constructive feedback has significantly improved the clarity, theoretical rigor, and reproducibility of our work. Below, we summarize the major changes and responses to the common concerns raised across the reviews.

**1. Theoretical Rigor and Mathematical Definitions (Reviewers 1, 3, 4)**
> **Concern:** Questions regarding the precise mathematical formulation of the ranking losses (PSR/LGR), the symmetry of pairwise weights, and the interpretation of theoretical bounds (Theorem 3.2).
>
> **Response:** We have revised Section 3 to rigorously define the PSR/LGR losses, ensuring notation consistency (e.g., symmetric weights $\lambda_{ij}$, iteration over unordered pairs). We also clarified that Theorem 3.2 bounds the *expected regression risk* of the combined objective, demonstrating that constant trivial predictors cannot minimize the ranking loss.

**2. Reproducibility and Code Availability (Reviewers 1, 3)**
> **Concern:** The need for open access to code, configuration files, and data splits to ensure the results are reproducible.
>
> **Response:** We have released a comprehensive anonymized repository (https://anonymous.4open.science/r/del_ranking_2333) containing full source code, configuration files with random seeds for all experiments, and the exact train/validation/test split indices used to generate the reported results.

**3. Practical Efficiency, Scalability, and Hyperparameters (Reviewers 1, 2, 4)**
> **Concern:** Inquiries about the wall-clock runtime, computational cost compared to baselines, and sensitivity to hyperparameters.
>
> **Response:** We added benchmarks (Appendix D.4) showing that DEL-Ranking scales linearly with library size and offers inference throughput comparable to MLP-ZIP and DEL-Dock on standard hardware. Furthermore, extensive ablations demonstrate that the model performance remains stable across a broad range of hyperparameter settings ($\rho, \gamma, \tau$).

**4. Role of CRC and Functional-Group Labels (Reviewers 2, 4)**
> **Concern:** The dependence of the model on functional-group labels and the applicability of the CRC module when such labels are unavailable.
>
> **Response:** We clarified that CRC is an optional, additive module; the model functions effectively using only Ranking+ZIP on label-free datasets (e.g., 5fl4). When enabled, CRC relies on coarse, easily automated substructure queries (e.g., presence of a sulfonamide motif) rather than expensive or unavailable annotations.

**5. Validation and Data Leakage (Reviewers 1, 4)**
> **Concern:** Clarification on data split strategies to avoid leakage and the external validation of discovered hits.
>
> **Response:** We confirmed that our evaluation on ChEMBL is strictly zero-shot (disjoint molecules from training), and internal DEL splits are random and fixed. We also highlighted orthogonal wet-lab validation, where DEL-Ranking identified novel pyrimidine sulfonamide hits with $K_i$ values comparable to canonical binders.

Best regards,

Authors

---

### Author Response · Authors · 2025-11-28
**Final Response to the Area Chair (1/2)**

Dear Area Chair,

Thank you for taking the time to evaluate our submission and the accompanying author responses following the recent re-assignment of area chairs. In this note, we summarize the main contributions of the work, explain how we have addressed the reviewers’ primary concerns, and clarify why we believe the paper meets the bar for acceptance.

We also confirm that we have strictly adhered to the ICLR review policies and code of conduct and have not engaged in any off-platform communication with reviewers.

---

## Summary of Contributions and Strengths

Our submission introduces **DEL-Ranking**, a DEL-specific framework that integrates a dual-perspective ranking objective (Pairwise Soft Rank, PSR, and Listwise Global Rank, LGR) with a ZIP-based count model and an optional Chemical-Referenced Correction (CRC) module. The framework is designed to address two central challenges in DNA-encoded library (DEL) screening:

1) distribution noise and zero-inflation in low-copy regimes, and
2) systematic shift between observed read counts and true binding affinities.

Across multiple carbonic anhydrase datasets, DEL-Ranking consistently improves ranking metrics (Spearman and subset Spearman) over strong baselines such as DEL-Dock, MLP-ZIP, and DEL-QSVR, with particularly pronounced gains in high-noise conditions (e.g., 4kp5-OA). Beyond numerical performance, the model recovers both the canonical benzene-sulfonamide scaffold and a previously underappreciated pyrimidine-sulfonamide motif, supported by orthogonal biochemical assays and structural analysis.

The work further contributes three DEL datasets combining 2D structures, 3D poses, and functional labels, together with full code and configuration files, thereby enabling reproducibility and follow-up research by the community.

---

## Resolution of Reviewers’ Main Concerns

**Theoretical Clarity and Loss Definitions**

Several reviewers requested more precise mathematical definitions of PSR and LGR, a clearer treatment of symmetric pairwise weights, and a more careful interpretation of Theorem 3.2.

> In the revised manuscript, Section 3 has been rewritten to provide rigorous and self-consistent definitions of the ranking losses, to state explicitly that the summation is over unordered pairs, and to clarify the role of the top-K normalization.

We also clarified that Theorem 3.2 concerns the **expected regression risk** of the combined objective
$L_C = \alpha L_{\text{ZIP}} + (1-\alpha)L_{\text{rank}}$,
and that the presence of a non-trivial ranking term prevents constant predictors from minimizing this combined loss. The assumptions and scope of the theorem are now explicitly stated, and the connection to standard learning-to-rank theory (e.g., NDCG-style surrogates as in LambdaRank/ListMLE) is made more precise.

Notational inconsistencies between equations and algorithmic descriptions have been resolved, and we explain how the gradients correspond to standard NDCG-style objectives. We believe that the theoretical exposition now meets the level of clarity and rigor expected for a conference paper.

**Reproducibility, Splits, and Data Leakage**

Reviewers emphasized the importance of transparent split strategies, absence of leakage, and availability of code and data.

> We have released an anonymized public repository that includes the full training and inference code, configuration files with all random seeds, and the exact train/validation/test split indices for each DEL dataset, along with scripts and instructions to reproduce all tables and figures.

In the main text, we now clearly state that DEL-Ranking is trained on DEL read counts and evaluated **zero-shot** on external ChEMBL datasets with **no overlap** in molecular identities between training libraries and evaluation ligands. Within each DEL dataset, molecule-level random splits with fixed ratios and a fixed random seed are used consistently across all baselines. These clarifications directly address concerns about data leakage and reproducibility.

---

### Author Response · Authors · 2025-11-28
**Final Response to the Area Chair (2/2)**

**Runtime, Scalability, and Hyperparameter Sensitivity**

There were questions regarding the practical runtime, scalability to realistic library sizes, and sensitivity to hyperparameters.

> We added runtime benchmarks (Appendix D.4) reporting training time per epoch and inference throughput (per 100k compounds) for DEL-Ranking, DEL-Dock, and MLP-ZIP under the same hardware setting.

These results show that DEL-Ranking scales linearly with the number of compounds and achieves training and inference costs comparable to commonly used baselines. Furthermore, we provide systematic ablations over key hyperparameters (ranking weight $\rho$, CRC weight $\gamma$, temperature $T$, margin $\tau$, top-K, and structure/fingerprint balance). The performance varies smoothly and remains strong over a broad range of settings, rather than depending on a narrow optimum.

We also describe a simple heuristic for choosing $\rho$ and $T$ by matching gradient magnitudes between $L_{\text{ZIP}}$ and $L_{\text{rank}}$ on a small validation batch. Overall, the method is practically implementable with standard computing resources and does not require delicate hyperparameter tuning.

**Role of CRC and Applicability to Label-Free Datasets**

Reviewers raised concerns about the dependency of CRC on functional-group labels and its applicability when such labels are unavailable.

> CRC is an optional component. On label-free datasets such as 5fl4, we set its weight $\gamma = 0$, so the model reduces to a ranking + ZIP architecture, which already outperforms DEL-Dock and related baselines.

When functional-group labels are available, they are constructed using simple, target-specific substructure rules (for example, the presence of the arylsulfonamide building block BB3-197 in CAS-DEL CA2/CA12), rather than relying on model-derived scores. This design limits self-confirmation bias and reinforces the interpretability of CRC as a chemically grounded refinement rather than a hard requirement for using DEL-Ranking.

**Experimental Validation and Interpretability**

Reviewers also asked whether the observed gains translate into meaningful chemical insight and practical value beyond purely numerical metrics.

> We clarified that the pyrimidine-sulfonamide motif identified by DEL-Ranking is supported by independent biochemical $K_i$ measurements for prioritized hits, as well as docking and interaction analyses showing coordination patterns consistent with, but distinct from, the canonical benzene-sulfonamide scaffold.

Subset analyses (such as SubSp metrics) and functional-group-level interpretations arise naturally from the architecture and loss design, helping to prioritize chemotypes in DEL-based hit discovery campaigns. These aspects strengthen the case that DEL-Ranking is not only a performance improvement but also a tool for interpretable and actionable decision-making.

---

**Overall Assessment**

Taking the reviews, rebuttal, and revisions together, we believe the paper now presents:

> (i) a DEL-specific methodological advance that unifies ZIP modeling, local and global ranking losses, and an optional chemically grounded correction term,
> (ii) a theoretically well-motivated and internally consistent objective,
> (iii) reproducible experiments on realistic DEL datasets with carefully controlled splits and open-source code and configurations, and
> (iv) practically relevant, interpretable insights into binding motifs, supported by orthogonal biochemical and structural evidence.

While the framework is more sophisticated than some baselines, this complexity is directly motivated by DEL-specific statistical challenges and is empirically justified by improved robustness under high-noise conditions and more informative motif discovery. We respectfully submit that these factors, taken together, meet the standard for acceptance at ICLR.

Thank you again for your time and careful consideration.

Sincerely,
Authors

---

### Note · Program_Chairs · 2026-01-17
**Submission Desk Rejected by Program Chairs**

The following references in this submission do not refer to real documents and/or have major errors in bibliographic information:

 Jonathan M Stokes, Kevin Yang, Kyle Swanson, Wengong Jin, Andres Cubillos-Ruiz, Nina M Donghia, Craig R MacNair, Shawn French, Lindsey A Carfrae, Zohar Bloom-Ackermann, et al. Deep learning-based prediction of protein-ligand interactions. Proceedings of the National Academy of Sciences, 117(32):19338-19348, 2020.